# Prompt Estimation from Prototypes for Federated Prompt Tuning of Vision Transformers

**M Yashwanth**[*]                                               *yashwanthm@iisc.ac.in*
*Department of Computational and Data Sciences, Indian Institute of Science.*

**Sharannya Ghosh**[*†]                                          *sharannyaghosh31@gmail.com*
*Accenture, Japan.*

**Aditay Tripathi** [‡]                                          *aditaytr@gmail.com*
*Google, India.*

**Anirban Chakraborty**                                          *anirban@iisc.ac.in*
*Department of Computational and Data Sciences, Indian Institute of Science.*

**Reviewed on OpenReview:** *https://openreview.net/forum?id=gO1CpPRj6A*

## Abstract

Visual Prompt Tuning (VPT) of pre-trained Vision Transformers (ViTs) has proven highly effective as a parameter-efficient fine-tuning technique for adapting large models to downstream tasks with limited data. Its parameter efficiency makes it particularly suitable for Federated Learning (FL), where both communication and computation budgets are often constrained. However, global prompt tuning struggles to generalize across heterogeneous clients, while personalized tuning overfits to local data and lacks generalization. We propose PEP-FedPT (Prompt Estimation from Prototypes for Federated Prompt Tuning), a unified framework designed to achieve both generalization and personalization in federated prompt tuning of ViTs. Within this framework, we introduce the novel Class-Contextualized Mixed Prompt (CCMP) — based on class-specific prompts maintained alongside a globally shared prompt. For each input, CCMP adaptively combines class-specific prompts using weights derived from global class prototypes and client class priors. This approach enables per-sample prompt personalization without storing client-dependent trainable parameters. The prompts are collaboratively optimized via traditional federated averaging technique on the same. Comprehensive evaluations on CIFAR-100, TinyImageNet, DomainNet, and iNaturalist datasets demonstrate that PEP-FedPT consistently surpasses the state-of-the-art baselines under diverse data heterogeneity scenarios, establishing a strong foundation for efficient and generalizable federated prompt tuning of Vision Transformers.

## 1 Introduction

Federated learning (FL) (McMahan et al., 2017) is a collaborative machine learning approach in which a central server coordinates multiple clients to jointly train a global model, while preserving the privacy of the client's data by keeping the local data decentralized. A major challenge in FL is data heterogeneity: datasets on each client can differ significantly in distribution, resulting in non-identically distributed (non-iid) data across the network. These discrepancies often cause client models to converge to different local minima, a phenomenon known as "client drift" (Karimireddy et al., 2020), which in turn degrades the generalization performance of the global model. To address this issue, personalized FL methods (Chen & Chao; Ma et al., 2022; Shamsian et al., 2021) have been proposed, aiming to better accommodate diverse local data distributions. However, a key drawback of personalized methods is their reduced generalization performance on new or unseen clients (Deng et al., 2024).

---

[*]Equal contribution.
[†]Work done during internship at Indian Institute of Science.
[‡]Provided valuable insights that helped this work.

Inspired by their success in centralized learning, large foundation models (FMs) are increasingly adopted in FL to mitigate data heterogeneity (Bommasani et al., 2021; Dosovitskiy et al., 2021; Radford et al., 2021). FMs demonstrate enhanced robustness in non-iid data settings (Qu et al., 2022). Yet, their substantial computational demands for tuning on resource-constrained edge devices and high communication costs present major hurdles. Parameter-efficient tuning methods, such as Visual Prompt Tuning (VPT) (Jia et al., 2022), offer a promising solution for efficient FM adaptation within FL, significantly reducing communication overhead while harnessing the power of large models.

Prompt tuning has recently gained attention in Federated Learning (FL), with methods like FedPR (Feng et al., 2023) and pFedPG (Yang et al., 2023) exploring its potential. However, each comes with notable limitations. FedPR employs global prompts and is primarily suited for cross-silo FL. This design struggles in highly heterogeneous client settings, where shared global prompts fail to adapt to diverse local data distributions (Li et al., 2020; Deng et al., 2024). pFedPG, on the other hand, generates client-specific prompts at the server and sends them to clients for local fine-tuning. Although personalized, this approach implicitly assumes full client participation in every round—an impractical assumption in many real-world FL systems. Moreover, such personalized strategies risk fitting to local data (Wu et al.; Deng et al., 2024), limiting their generalization to unseen or non-participating clients (as seen in Table 3). To address these issues, SGPT (Deng et al., 2024) employs shared and group-specific prompts to enhance generalization. However, its two-stage training process and non-differentiable mechanism add optimization complexity and computational overhead. Furthermore, when data heterogeneity is high, it still struggles to generalize across diverse client distributions (Tables 1 and 2). These limitations highlight a key trade-off in FL prompt tuning: global prompts generalize well but lack expressiveness, while personalized prompts offer local adaptability but suffer from poor generalization and scalability. This raises the central question:

*Can we achieve effective personalization while relying solely on globally shared prompts?*

We answer this by proposing a novel prompt-tuning strategy tailored for fine-tuning of vision transformer in FL. Our method introduces class-specific prompts that are jointly optimized with shared prompts to address data heterogeneity across clients. To induce personalization without local prompt storage, we propose Prompt Estimation from Prototypes for Federated Prompt Tuning (PEP-FedPT). It generates a Class-Contextualized Mixed Prompt (CCMP) by combining global class-specific prompts. The combination weights are determined by per-class membership scores, computed using global `cls`-token prototypes and the client's local class priors. The global `cls` prototype aggregates class centroids across clients, where each centroid is computed by averaging the `cls` token representations of data points within a given class. For a given input, we estimate class membership scores—refined by each client's class priors. These scores act as soft weights to combine class-specific prompts into a single, differentiable mixed prompt (CCMP). Owing to strong semantic structure learned by pre-trained vision transformers, samples from the same class tend to form compact clusters in the representation space. This property allows the global class centroids to serve as reliable anchors for estimating class membership under domain heterogeneity. Incorporating class priors accounts for label heterogeneity across clients: clients may observe the same label set but with markedly different class frequencies. Local priors reflect the client's underlying data distribution. Together, the similarity scores and class priors provide a robust estimate of class membership that captures both domain and label heterogeneity. Each client can dynamically personalize prompts using only shared global information, without the need for local prompt storage or specialized server-side generation. Our approach integrates seamlessly into standard FL pipelines and delivers strong empirical performance across heterogeneous datasets, as demonstrated in Tables 1 and 2. Despite relying solely on global prompts, it effectively utilizes clients' data distribution, achieving scalable generalization. Since early layers lack abstraction, while late layers limit prompt influence, CCMP is injected at intermediate layers, where representations are sufficiently semantic yet allow adequate depth for effective optimization. Our key contributions are as follows:

1. We introduce a unified framework PEP-FedPT that jointly optimizes class-specific and shared prompts to address data heterogeneity in federated learning of ViTs. We exploit the clients' distribution to achieve personalization by using only global prompts. Thus our proposed strategy aims to strike an effective balance between generalizability and personalization.

2. We design a novel prompt-mixing strategy that generates Class Contexualized Mixed Prompts (CCMP) as a function of class-specific prompts shared globally across clients. We empirically demonstrate its

superiority over existing methods through extensive experiments on datasets exhibiting feature and label imbalance.

3. We provide theoretical insights into our design by showing that CCMP minimizes a quadratic upper bound and is optimal in the Minimum Mean Squared Error (MMSE) sense.

## 2 Related Work

### 2.1 Federated Learning (FL):

FL is a machine learning paradigm that emphasizes data privacy by enabling collaborative model training under the coordination of a central server, without requiring direct data sharing. FedAvg (McMahan et al., 2017) is the most commonly used technique for aggregating local models. This has led to its broad adoption across various domains, such as Internet of Things and mobile devices (Mills et al., 2019; Nguyen et al., 2021; Hard et al., 2018; Ramaswamy et al., 2019), healthcare (Rieke et al., 2020; Xu et al., 2021; Nguyen et al., 2021; Brisimi et al., 2018; Feng et al., 2023), person re-identification (Zhuang et al., 2020), and face recognition (Liu et al., 2022a). Under data-heterogeneity, training the models using FedAvg leads to client-drift . Addressing this, regularization techniques (Acar et al.; Li et al., 2020; Gao et al., 2022) and variance reduction methods (Karimireddy et al., 2020), and several studies on improving the generalization performance in FL by inducing flatness during the local training (Sun et al., 2023; Caldarola et al., 2022) have come to light. In case of extreme data heterogeneity across the clients, training a single model for all the clients will be difficult. As an alternative, personalized FL approaches have been proposed (Tan et al., 2023; Chen & Chao; Ma et al., 2022; Shamsian et al., 2021). These frameworks share some model parameters with the server while keeping others client-specific, enabling adaptation to local data distributions. Federated Domain Generalization aims to learn models robust to client-level domain shifts; representative approaches include FedSR Nguyen et al. (2022), GA Zhang et al. (2023), and FedGaLA Pourpanah et al. (2025), which address this via representation regularization, variance-aware aggregation, and gradient alignment, respectively.

### 2.2 Prompt Tuning and Federated Learning:

As the fine-tuning of the foundational models became ubiquitous for downstream tasks in centralized learning (Bommasani et al., 2021; Dosovitskiy et al., 2021; Radford et al., 2021), prompt tuning techniques were originally proposed in the NLP community (Li & Liang, 2021; Liu et al., 2022b). Recently, Visual Prompt Tuning (VPT) was proposed for prompt tuning in ViT models, demonstrating its efficiency. ViT based FL (Qu et al., 2022) shows robustness to heterogeneity. However, due to heavy communication costs, prompt-tuning on pre-trained VIT models gained attention. In the recent literature, prompt tuning for FL has been proposed in methods like SGPT (Deng et al., 2024), FedPR (Feng et al., 2023), pFedPG (Yang et al., 2023) and also in the context of Vision and Language models FedOTP (Li et al., 2024). SGPT relies on group-specific prompts and requires alternate training due to a non-differentiable selection mechanism. pFedPG depends on full client participation, while FedOTP assumes access to both text and image encoders. We address these limitations by introducing class-specific prompts and a novel per-sample prompt-mixing strategy based on class priors and `cls`-token prototypes. Jin et al. (2024) explore prompt-based domain adaptation by tuning prompts in pre-trained vision–language models for cross-domain transfer.

## 3 Preliminary

In this paper, we use boldface letters to denote matrices and vectors. The operator "$\cdot$" refers to element-wise multiplication and $*$ refers to the standard matrix multiplication. We define $TL_i$ as the $i^{th}$ transformer layer, and $\mathbb{E}$ denotes the expectation operator. The term $\delta_k^c$ represents the probability of observing samples from class $c$ at client $k$. Additionally, $sim(\mathbf{p}, \mathbf{q})$ denotes the cosine similarity, while $\mathbb{I}$ represents the indicator function. $[M]$ denotes the set $\{1, 2, ., .M\}$[1].

### 3.1 Visual Prompt Tuning (VPT)

VPT is a parameter-efficient fine-tuning method for the pre-trained ViT (Jia et al., 2022). It is an efficient alternative to fine-tuning the full model. Jia et al. (2022) propose VPT, where prompts are inserted at the input of the ViT. with $d$ dimensional trainable prompt $\mathbf{P}_0 \in \mathbb{R}^{d \times 1}$ as follows:

$$\mathbf{cls}_i, \mathbf{P}_i, \mathbf{E}_i = TL_i([\mathbf{cls}_{i-1}, \mathbf{P}_{i-1}, \mathbf{E}_{i-1}]), \tag{1}$$

---

[1]The detailed notations and definitions are in Sec. A.1 of Appendix

$$\mathbf{y} = \mathbf{H} * \mathbf{cls}_M. \tag{2}$$

$\mathbf{E_i} \in \mathbb{R}^{d \times n_I}$ denotes the image tokens at layer $i$, $n_I$ denotes the number of image tokens, $M$ denotes the number of layers in the transformer. The final layer's $\mathtt{cls}$ token i.e, $\mathbf{cls}_M$ is used for classification. In the above model, the classification head $\mathbf{H}$ and $\mathbf{P}_0$ are trainable.

## 3.2 Federated Learning (FL)

In FL, the server orchestrates the training with $n$ clients with the goal of minimizing the following training objective:

$$\min_{\boldsymbol{\theta}} f(\boldsymbol{\theta}) \coloneqq \frac{1}{n} \sum_{k=1}^{n} f_k(\boldsymbol{\theta}), \tag{3}$$

$f_k$ denotes the $k^{th}$ client local objective function. $\boldsymbol{\theta}$ denotes the model parameters shared across the clients. In general, it can be written as $f_k(\boldsymbol{\theta}) = \underset{(\mathbf{x},y) \sim \mathcal{D}_k}{\mathbb{E}} l_k(\boldsymbol{\theta}; (\mathbf{x}, y))$. $\mathcal{D}_k$ denotes the data distribution of the client $k$ and $l_k(\boldsymbol{\theta}; (\mathbf{x}, y))$ denotes the task-specific loss function. For a classification task, $\mathbf{x}$ denotes the input and $y$ is the ground truth. In FL training, at each round $t$, the server broadcasts the global model $\boldsymbol{\theta}^t$ to a randomly selected subset of clients $S_t$. Each client $k \in S_t$ performs several steps of local training starting from $\boldsymbol{\theta}^t$, and then sends its updated model $\boldsymbol{\theta}_k^t$ back to the server. The server aggregates these updates using federated averaging:

$$\boldsymbol{\theta}^{t+1} = \frac{1}{|S_t|} \sum_{k \in S_t} \boldsymbol{\theta}_k^t.$$

The updated model $\boldsymbol{\theta}^{t+1}$ is then broadcast to clients in the next round. This procedure describes the basic FedAvg algorithm McMahan et al. (2017).

# 4 Proposed Method: Prompt Estimation from Prototypes- Federated Prompt Tuning (PEP-FedPT)

We consider a federated learning setup with $n$ clients coordinated by a central server, where each client's data is drawn from a distinct distribution $\mathcal{D}_k$. Following VPT (Jia et al., 2022), we assume each client uses a pre-trained ViT-B/16 as its local model architecture. We introduce Shared Prompts and Class-Contextualized Mixed Prompts (CCMP). Also, by utilizing the information in the local class priors and the global class prototypes, we softly combine the class-specific prompts, leading to per-client customization while sharing the global class-specific prompts. A highlevel overview of our method is shown in Algorithm.1.

## 4.1 Prompt Design

Upon insertion of prompt $\mathbf{P}_{S_{l-1}}$ into layer $l$ of a ViT, the input and output for that layer can be written as:

$$\mathbf{cls}_l, \mathbf{P}_{S_l}, \mathbf{E}_l = TL_l(\mathbf{cls}_{l-1}, \mathbf{P}_{S_{l-1}}, \mathbf{E}_{l-1}), \tag{4}$$

where $\mathbf{cls}_{i'}$ denotes the cls-token representation at output layer $i'$ of ViT, and $\mathbf{E}_{i'}$ denotes the combined representation of the remaining tokens.

We have two sets of trainable parameters: Shared Prompts $\mathbf{P}_S$ and Class-Specific Prompts $\mathbf{P}_C$, both of which are common across all clients. While we insert the shared prompts directly, the Class-Specific Prompts are used to compute Class Contexualized Mixed Prompts (CCMP) (denoted by $\mathbf{m}$) , which are then inserted in subsequent layers in the ViT. We now describe each of these in detail.

**Shared Prompts** ($\mathbf{P}_S$): Inspired by Jia et al. (2022) we added shared prompts at the very first layer of the ViT model. These are shared across the clients and are given by $\mathbf{P}_S = \left[\mathbf{p}_{s_1}\mathbf{p}_{s_2}...\mathbf{p}_{s_{|S|}}\right]$, where $|S|$ is the number of shared prompts inserted and are processed as follows:

$$\mathbf{cls}_1, \mathbf{P}_{S_1}, \mathbf{E}_1 = TL_1([\mathbf{cls}_0, \mathbf{P}_S, \mathbf{E}_0]). \tag{5}$$

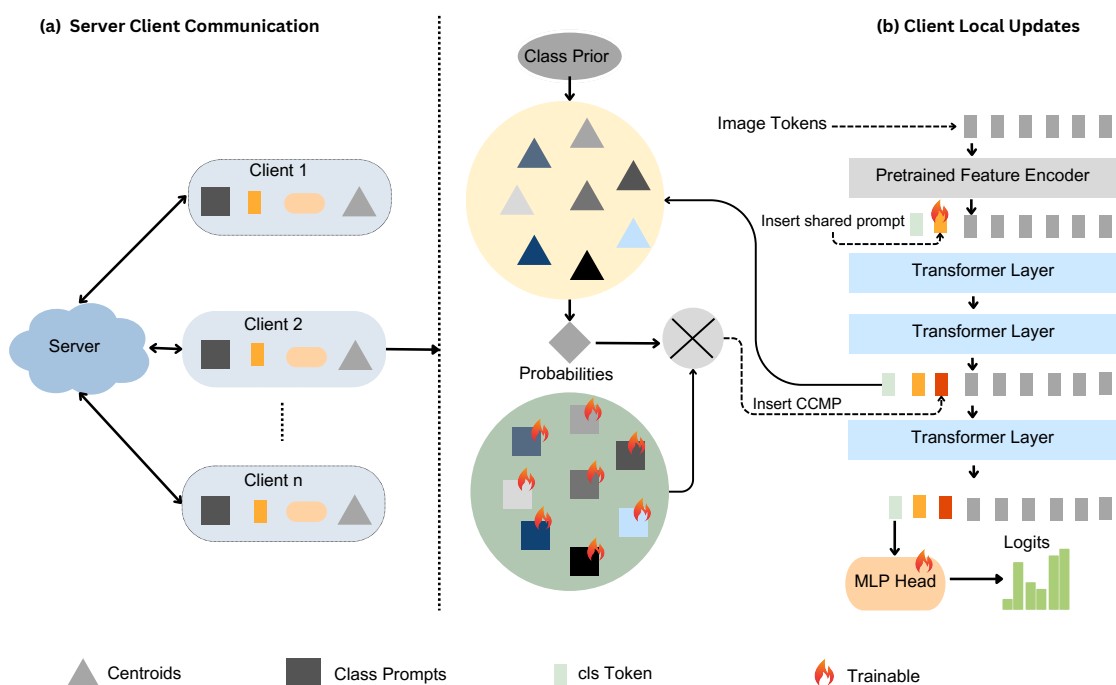

Figure 1: The left panel (a) illustrates server-client communication during federated training. In each communication round, clients (right panel (b)) insert shared prompts at the input of the transformer and class-contextualized prompts— derived by mixing class prompts using probabilities computed from local class priors, cls-tokens and centroids—at intermediate layer(s).

---

**Algorithm 1:** Pseudocode for PEP-FedPT

1  Compute global class prototypes before training;
2  **for** *each communication round* **do**
3      Server sends model parameters to clients;
4      **for** *each client in parallel* **do**
5          **for** *local epochs* **do**
6              Insert shared prompts at layer 0;
7              **for** *layers in which CCMP is inserted* **do**
8                  Compute class scores from priors and CLS-prototype similarity;
9                  Mix scores with class prompts to form CCMP;
10                 Insert CCMP;
11             Train shared and class prompts;
12         Send prompt updates and class prototypes to server;
13     Server aggregates prompt updates;
14 Periodically, server aggregates class prototypes;

---

As discussed in Ostapenko et al. (2022) and Deng et al. (2024), the early layers capture the low-level representations and can be shared across the classes. This allows the model to have better generalization. It is shown that the representations in the initial layer of pre-trained ViT are uniformly distributed on the manifold, indicating that the information is shared across the classes. This is true even when the data distribution is different across the clients as shown Sec A.5.4 of the appendix.

**Class Contextualized Mixed Prompts** (CCMP) ($\mathbf{m}(k)$): This prompt is obtained by softly combining the class-specific prompts given by $\mathbf{P}_C = \left[\mathbf{p}_{c_1}\mathbf{p}_{c_2}...\mathbf{p}_{c_{|C|}}\right]$ with scores driven by client-specific data distribution, where $|C|$ is equal to the total number of classes and $\mathbf{P}_C \in \mathbb{R}^{d \times |C|}$. These class-specific prompts $\mathbf{P}_C$ are shared across the clients, but the scores that act as weights to combine these class-specific prompts are local to each client. These soft weights for a client $k$ at the input of layer $l$ on the $i$-th training input denoted by $\mathbf{s}_{i,l-1,k} \in [0,1]^{|C| \times 1}$ are designed as the function of input data point $\mathbf{x}_{i,k}$, cls token prototypes and class priors. Finally, the CCMP $\mathbf{m}_{l-1}$ is added at the input of layer $l$, and it's given below

$$\mathbf{m}_{l-1}(k) = \mathbf{P}_C * \mathbf{s}_{i,l-1,k}. \tag{6}$$

The overall input and output after adding the CCMP $\mathbf{m}_{l-1}$ at the input of layer $l$ is shown below:

$$\mathbf{cls}_l, \mathbf{m}_l(k), \mathbf{P}_{S_l}, \mathbf{E}_l = TL_l([\mathbf{cls}_{l-1}, \mathbf{m}_{l-1}(k), \mathbf{P}_{S_{l-1}}, \mathbf{E}_{l-1}]). \tag{7}$$

The soft weights are explained in Sec. 4.2. In Table 13, we show that inserting CCMP prompts too early in the ViT is not advantageous, as the cls token representations in the initial layers are not sufficiently informative for reliable estimation of CCMP scores. Conversely, adding prompts in later layers is also suboptimal: although the cls token representations are stronger at these stages, the prompts are not inserted deeply enough to learn meaningful adaptations. Therefore, the most effective placement is within the mid-to-late layers (namely 5, 6 and 7 for all our experiments), where the representations are both informative and feasible for effective prompt learning.

If the ViT has $M$ layers, the final logits are given by:

$$\mathbf{y} = \mathbf{H} * \mathbf{cls}_M. \tag{8}$$

$\mathbf{H}$ denotes the classification layer. Finally, we aim to solve the following federated optimization problem involving the shared and class-specific prompts.

$$\min_{\mathbf{P}_S, \mathbf{P}_C, \mathbf{H}} \frac{1}{n} \sum_{k=1}^{n} f_k(\mathbf{w}_{pre}; \mathbf{P}_S, \mathbf{P}_C, \mathbf{H}). \tag{9}$$

Here $\mathbf{w}_{pre}$ denotes the pre-trained ViT parameters and $f_k$ denotes the loss for the client $k$.

## 4.2 Estimation of Soft Weights for CCMP

We present the design of the soft weights, which are aimed to provide class-specific information to the model. We exploit the information present in the class prototypes of $\mathbf{cls}_{l-1}$ token at a layer $l$. Our empirical observations show that cls tokens of the pre-trained ViT model carry significant information regarding the downstream task. In the Figure 2 we observe the Top-5 zero-shot test accuracy of the CIFAR-100 dataset computed at each layer. The accuracy at layer $l$ is computed by taking the minimum distance between the cls token corresponding to the test input and the class prototypes of the cls token at the input of layer $l$.

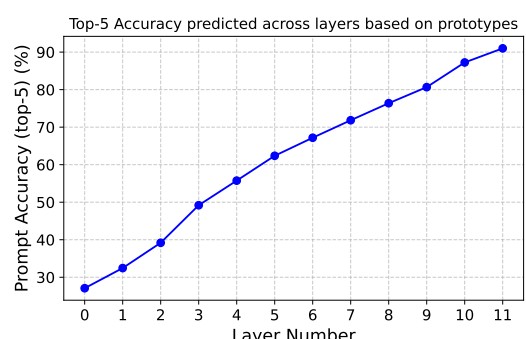

We now describe the soft weights computed for a layer $l$ i.e, $\mathbf{s}_{l-1}$. Let us denote the cls token at the input of layer $l$ corresponding to data point $\mathbf{x}_{i,k}$ for client $k$ in communication round $t$ as $\mathbf{cls}_{l-1,i,k,t}$. The cls token's class prototype for the class $c$ at communication round $t$ is denoted by $\boldsymbol{\mu}_{l-1,k,t}^c$ and it is computed as in Eq. 10

$$\boldsymbol{\mu}_{l-1,k,t}^c = \begin{cases} \frac{1}{n_{k,c}} \sum_{i=1}^{N_k} \mathbf{cls}_{l-1,i,k,t} \cdot \mathbb{I}_{y_{i,k}=c}, & n_{k,c} > 0, \\ \mathbf{0}, & n_{k,c} = 0. \end{cases} \tag{10}$$

Figure 2: The Top-5 accuracy computed based on the minimum distance between the cls token corresponding to the input and the cls prototypes. This shows that the cls representations in the middle layers have coarse information of the task.

Here $n_{k,c} = \sum_{i=1}^{N_k} \mathbb{I}_{y_{i,k}=c}$, where $N_k$ denotes the number of
data points of client $k$, and $\mathbb{I}_{y_{i,k}=c}$ denotes indicator function. It takes value 1 if the data point $i$ of client $k$
belongs to class $c$ otherwise it is 0.

After every fixed update period $R$, the server aggregates the prototypes from the clients to compute the
aggregated prototype $\hat{\boldsymbol{\mu}}_{l-1,r}^c$ at the $r$-th period as Eq. 11. Let the set of communication rounds within this
update period be $\Lambda = \{rR, rR+1, \ldots, (r+1)R-1\}$.

$$\hat{\boldsymbol{\mu}}_{l-1,r}^c = \begin{cases} \frac{1}{D_c} \sum_{t \in \Lambda} \sum_{k \in S_t} \mathbb{I}_{\{\boldsymbol{\mu}_{l-1,k,t}^c \neq \mathbf{0}\}} \boldsymbol{\mu}_{l-1,k,t}^c, & D_c > 0, \\ \\ \mathbf{0}, & D_c = 0. \end{cases} \tag{11}$$

here $D_c = \sum_{t \in \Lambda} \sum_{k \in S_t} \mathbb{I}_{\{\boldsymbol{\mu}_{l-1,k,t}^c \neq \mathbf{0}\}}$, $S_t \subseteq [n]$ denotes the subset of clients sampled by the server at a round
$t$. As the training progresses, the server uses the momentum to update the aggregated prototypes $\hat{\boldsymbol{\mu}}_{l-1,r}^c$ to
form the global class prototype $\boldsymbol{\mu}_{l-1,r}^c$ as in Eq. 12 which is then communicated to the clients. The parameter
$\rho$ denotes the momentum. The updated prototypes $\boldsymbol{\mu}_{l-1,r}^c$ are sent to the clients.

$$\boldsymbol{\mu}_{l-1,r}^c = \rho \cdot \boldsymbol{\mu}_{l-1,r-1}^c + (1-\rho) \cdot \hat{\boldsymbol{\mu}}_{l-1,r}^c. \tag{12}$$

If $D_c$ is 0, we set $\rho = 1$.

We now define the un-normalized score function $\hat{s}_{i,l-1,k}^c$ assigned by the cls token $\mathbf{cls}_{l-1,i,k}$, corresponding to
input $\mathbf{x}_{i,k}$, to class-specific prompt $\mathbf{p}_c$ at the client $k$. Here we drop the index of the communication round $t$
and the update period $r$ for better readability.

$$\hat{s}_{i,l-1,k}^c = exp\left(\frac{sim\left(\mathbf{cls}_{l-1,i,k}, \boldsymbol{\mu}_{l-1}^c\right)}{\tau}\right) \delta_k^c. \tag{13}$$

We define $sim(\mathbf{p}, \mathbf{q}) = \frac{\mathbf{p}^\top \mathbf{q}}{\|\mathbf{p}\|\|\mathbf{q}\|}$.

$\tau$ is the hyper-parameter and $\delta_k^c$ is the prior probability of the class $c$ at the client $k$. This can be obtained
by computing the empirical label distribution. Availability of such class prior information is a common
assumption in the works such as Lee et al. (2022)

$$\delta_k^c = \frac{1}{N_k} \sum_{i=1}^{N_k} \mathbb{I}_{y_{i,k}=c}. \tag{14}$$

The final scores $s_{i,l-1,k}^c$ are obtained as

$$s_{i,l-1,k}^c = \frac{\hat{s}_{i,l-1,k}^c}{\sum_{m=1}^{|C|} \hat{s}_{i,l-1,k}^m}. \tag{15}$$

The scores $s_{i,l-1,k}^c$ can be interpreted as the probability assigned to the class-specific prompt $\mathbf{p}_c$, given the
$\mathbf{cls}$ token $\mathbf{cls}_{l-1,i,k}$. All the probabilities across the classes form the desired weight vector $\mathbf{s}_{i,l-1,k}$ and the
final CCMP $\mathbf{m}_{l-1}$ is computed using Eq. 6. CCMP achieves personalization even when client priors are
uniform, due to its ability to capture the domain gap within the scores. This is a direct result of the $\mathbf{cls}_{l-1,i,k}$
tokens, as they will have different representations for different domains.

**Privacy considerations**: We acknowledge that PEP-FedPT requires the transmission of class prototypes to
facilitate adaptive prompt mixing. However, these transmitted statistics represent aggregated, intermediate
layers', low-dimensional summaries of the local data rather than raw data. Moreover, without formal privacy
protection measures, strict privacy guarantees are challenging in virtually any FL framework. As discussed in
the Sec A.6 of (Xu et al.), it is usually hard for the server to extract sensitive data from category-level feature
statistics (in our case cls-prototypes) alone without having access to the feature extractor layers. Prototype-
based learning in FL has been used previously in (Dai et al., 2023; Tan et al., 2022; Xu et al.). Moreover in
practical scenarios, even gradient-sharing algorithms like FedAvg would require a privacy preserving technique
for the system to be completely reliable. In A.4.1, we have discussed the impact of adding Laplace DP noise
to the class prototypes on the overall performance of our method. We have observed that it has had minimal
impact on the final accuracy of our proposed method. We also show that $(\epsilon, 0)$ privacy can be attained.

### 4.3 Theoretical Underpinnings of CCMP

#### 4.3.1 CCMP minimizes the Quadratic Upper bound on the Loss around the class prompts

CCMP constitutes the core component of our methodology and by design, it is specific to each client. We show that CCMP minimizes the quadratic upper bound on the loss. We denote the estimate of the class prompt for class $i$ in any round as $\mathbf{p}_{c_i}$ and the class prompts will be denoted by $\mathbf{P}_C = [\mathbf{p}_{c_1}, \mathbf{p}_{c_2} \ldots, \mathbf{p}_{c_{|C|}}]$. Let $\mathbf{m}(k)$ denote the CCMP prompt used for client $k$ [2], and let the total number of clients be $n$, and $\delta_k^i$ denote the empirical probability that a data point at client $k$ belongs to class $i$. Let $\mathcal{P}$ be the set of all possible prompts across all the clients, such that $\mathbf{m}(k) \in \mathcal{P}, \quad \forall k \in \{1, 2, \ldots, n\}$. We denote the average loss corresponding to a data point whose true label $y = i$ as $l^i$. We rewrite this loss in terms of prompt $\mathbf{p}$ and class $i$ as $l^i(\mathbf{p})$. Then the global loss across all clients can be computed as $f := \frac{1}{n} \sum_{k=1}^{n} \left[ \sum_{i=1}^{|C|} \delta_k^i \cdot l_k^i(\mathbf{m}(k)) \right]$. We now state the following assumptions:

**A 1.** $\mathcal{P}$ *is compact subset of* $\mathbb{R}^d$, *where $d$ is the token dimension.*
This means that all the possible values for the prompts lie within a bounded region and they do not drift off to infinity.

**A 2.** $l_k^i$ *is Lipschitz smooth with parameter* $\beta_i \ \forall i \in [|C|], \forall k \in [n].$
This implies that the gradients do not change abruptly.

**A 3.** $l_k^i(\mathbf{p})$ *achieves its' minimum value for* $\mathbf{p} = \mathbf{p}_{c_i}^*.$
Intuitively this assumption implies that clients class-specific loss function has same minimum. This is true for label imbalance and is not true for feature imbalance setting.

**Proposition 1.** *If the above assumptions hold, we show that $f$ can be upper bounded as $f \leq \tilde{L} = \frac{1}{n} \sum_{k=1,i=1}^{n,|C|} \delta_k^i \left( l_k^i(\mathbf{p}_{c_i}) + \frac{\beta_{\max}}{2} \|\mathbf{m}(k) - \mathbf{p}_{c_i}\|^2 \right) + \tilde{C}$ and it is minimized at $\mathbf{m}(k) = \sum_{i=1}^{|C|} \delta_k^i \mathbf{p}_{c_i}, \quad \forall k \in [n].$ which is equivalent to the (CCMP) described in sec.4.2 as $\tau >> 1$. $\beta_{\max} = \max_{i \in [|C|]} \beta_i$, $\tilde{C}$ is a constant which depends on $\mathcal{P}$. Under the assumption. 3 which correspond to label heterogeneity setting this vanishes when $\mathbf{p}_{c_i} = \mathbf{p}_{c_i}^* \forall i \in [|C|]$ which makes $\tilde{L}$ a tight upper bound of $f$.*

A detailed proof is provided in Section A.6.1 of the appendix. The proof sketch proceeds by constructing an upper bound on the class-wise loss using smoothness assumptions. Specifically, the first-order term is bounded via compactness and smoothness properties. The upper bound is then minimized with respect to each $\mathbf{m}(k)$. The key insight from this proposition is that each client's prompt differs due to client-specific class priors, even though the underlying class prompts $\mathbf{p}_{c_i}$ are shared globally. This mechanism enables the prompts to adapt to each client's data distribution. The mixing scores in Eq. 15 depend both on the data instance and the class priors; this is further discussed in Sec A.6.4 of appendix.

#### 4.3.2 CCMP is Optimal in Minimum Mean Squared Estimate (MMSE) Sense

We denote $p_k(\mathbf{p}|\mathbf{cls}_{l-1})$ as the posterior probability of the prompt $\mathbf{p}$ after observing the $\mathtt{cls}$ token $\mathbf{cls}_{l-1}$ at the input of layer $l$. [3] It should be noted that this is a discrete probability measure over the class-specific prompts $\{\mathbf{p}_{c_1}, \mathbf{p}_{c_2} \ldots, \mathbf{p}_{c_{|C|}}\}$. If we assume that the density over the $\mathtt{cls}$ tokens follows $p_k(\mathbf{cls}_{l-1})$ and the posterior over the class given the $\mathtt{cls}$ token is modeled as (based on Eq. 15) i.e.,

$$p_k(\mathbf{p} = \mathbf{p}_c | \mathbf{cls}_{l-1}) = s_{i,l-1,k}^c. \tag{16}$$

This induces the joint probability density over the $\mathtt{cls}$ tokens observed and the class-specific prompts $\{\mathbf{p}_{c_1}, \mathbf{p}_{c_2} \ldots, \mathbf{p}_{c_{|C|}}\}$. We denote this by $p_k(\mathbf{cls}_{l-1}, \mathbf{p})$ and is given below

$$p_k(\mathbf{cls}_{l-1}, \mathbf{p}) = p_k(\mathbf{p}|\mathbf{cls}_{l-1})p_k(\mathbf{cls}_{l-1}). \tag{17}$$

**Proposition 2.** *If the cls tokens and the class-specific prompts at input of layer $l$ has the joint density given by $p_k(\mathbf{cls}_{l-1}, \mathbf{p})$ as in Eq. 17, then the CCMP prompt for a client $k$, $\mathbf{m}_{l-1}(k)$ obtained in Eq. 6 is Minimum Mean Squared Estimator (MMSE) of the true class prompt.*

---

[2]for notation convenience, we drop the layer index $j$ from $\mathbf{m}_j(k)$.
[3]In $\mathbf{cls}_{l-1}$ we omit the subscripts of client $k$, data point $i$ and round $t$ for simplifying notation.

The proposition 2 says that the CCMP obtained in Eq. 6 is optimal in MMSE sense. The detailed proof is given in Sec. A.6.2 of the appendix. This is done by showing MMSE optimality of the estimator $\mathbb{E}[\mathbf{p}|\mathbf{cls}_{l-1}]$. The discussion regarding the convergence is provided in Sec. A.6.3 of the appendix.

## 5 Experiments

**Datasets**: We conducted extensive experiments on four popular datasets (1) *CIFAR-100* (Krizhevsky & Hinton, 2009) dataset consists of 50,000 training images and $10,000$ test images distributed across 100 classes. (2) *TinyImageNet* (Le & Yang, 2015) contains $100K$ images of 200 classes, with each class containing 500 training images and 50 test images. (3) *DomainNet* (Peng et al., 2019) comprises 0.6 million images of 345 classes distributed across six domains: Clipart, Infograph, Painting, Quickdraw, Real, and Sketch; however, following the protocol of Yang et al. (2023), we use the top ten most frequent classes for our experiments. (4) *iNaturalist* originally introduced in Van Horn et al. (2018) is a large-scale fine-grained visual classification dataset comprised of images of natural species. In this work we use federated version of this dataset.

**Setup**: We split CIFAR-100 and Tiny-ImageNet datasets among the clients with two different settings of data heterogeneity: pathological splitting (Li et al., 2023; Oh et al.; Deng et al., 2024), where each client observes only 10 classes, and Dirichlet-based splitting denoted by $Dir(\xi)$ (Acar et al.), where each client has a non-identical label distribution. A lower $\xi$ value indicates higher heterogeneity, and we set $\xi = 0.3$. For CIFAR-100 we consider 100 clients and for TinyImageNet we consider 200 clients. Only 5 randomly chosen clients participate in every round. For the DomainNet dataset we consider the setting as Deng et al. (2024); Li et al. (2021) where each domain is allocated to 10 clients among 60 clients, this considers the scenario of feature imbalance setting. 6 randomly sampled clients participate in each communication round. Finally, in the iNaturalist dataset (Hsu et al., 2020), we make sure each client gets at least 16 training samples. This will have around $100k$ training samples distributed among the 1018 clients and 1203 classes. The partition is performed to mimic the cross-device (Kairouz et al., 2021) non-iid setting. In our method (PEP-FedPT) the CCMP is insterted at the layers $5, 6, 7$. In all the above experimental setups, the partition of the dataset across clients is completely disjoint, i.e., no two clients contain the same data sample in all cases. The visualization of data heterogeneity is provided in Sec. A.3.1 of appendix. The detailed hyperparameter settings is provided in A.3.2 and the evolution of cls representations are provided in the Sec. A.5.4 of the Appendix.

Table 1: Quantitative comparisons on CIFAR-100, Tiny-ImageNet datasets using ViT-B/16. We report the accuracy under two non-iid data partitioning setups 1) pathological: Each client observes only 10 classes 2) Dirichlet: Label distribution of each client is drawn from Dirichlet distribution.

| Datasets | CIFAR-100 (%) ↑ | | | | Tiny-ImageNet (%) ↑ | | | |
|---|---|---|---|---|---|---|---|---|
| Method | Pathological | | $Dir(0.3)$ | | Pathological | | $Dir(0.3)$ | |
| | Mean Acc | Worst Acc | Mean Acc | Worst Acc | Mean Acc | Worst Acc | Mean Acc | Worst Acc |
| Head-Tuning | $77.85_{\pm 0.17}$ | $59.87_{\pm 0.74}$ | $79.56_{\pm 0.25}$ | $66.66_{\pm 2.79}$ | $68.39_{\pm 0.76}$ | $44.09_{\pm 0.58}$ | $70.73_{\pm 0.08}$ | $45.63_{\pm 2.05}$ |
| FedVPT | $83.62_{\pm 0.02}$ | $70.19_{\pm 0.11}$ | $84.91_{\pm 0.07}$ | $74.64_{\pm 0.74}$ | $74.20_{\pm 0.33}$ | $54.00_{\pm 2.46}$ | $76.57_{\pm 0.34}$ | $50.34_{\pm 2.51}$ |
| FedVPT-D | $85.15_{\pm 0.77}$ | $70.12_{\pm 0.20}$ | $88.60_{\pm 0.19}$ | $79.17_{\pm 0.65}$ | $79.60_{\pm 0.42}$ | $59.83_{\pm 1.66}$ | $83.30_{\pm 0.16}$ | $60.33_{\pm 0.58}$ |
| FedPR | $81.77_{\pm 0.30}$ | $68.99_{\pm 0.48}$ | $82.27_{\pm 0.22}$ | $73.29_{\pm 1.38}$ | $68.86_{\pm 0.17}$ | $47.50_{\pm 1.63}$ | $68.93_{\pm 0.11}$ | $47.37_{\pm 1.44}$ |
| SGPT | $84.16_{\pm 0.24}$ | $70.79_{\pm 0.30}$ | $85.90_{\pm 0.21}$ | $76.73_{\pm 1.60}$ | $75.65_{\pm 1.81}$ | $55.66_{\pm 3.32}$ | $78.84_{\pm 1.11}$ | $53.87_{\pm 0.46}$ |
| pFedPG | $92.96_{\pm 1.34}$ | $84.58_{\pm 1.1}$ | $77.27_{\pm 0.77}$ | $62.34_{\pm 1.53}$ | $82.93_{\pm 0.18}$ | $50.21_{\pm 1.05}$ | $55.91_{\pm 0.65}$ | $49.31_{\pm 1.05}$ |
| P-PT | $75.37_{\pm 0.39}$ | $55.14_{\pm 1.03}$ | $80.10_{\pm 0.25}$ | $68.33_{\pm 0.58}$ | $61.68_{\pm 1.16}$ | $38.09_{\pm 6.38}$ | $62.78_{\pm 0.38}$ | $40.30_{\pm 1.13}$ |
| PEP-FedPT(Ours) | $\mathbf{95.46_{\pm 0.16}}$ | $\mathbf{84.74_{\pm 3.12}}$ | $\mathbf{88.75_{\pm 0.25}}$ | $\mathbf{81.00_{\pm 0.00}}$ | $\mathbf{91.52_{\pm 0.11}}$ | $\mathbf{77.33_{\pm 1.84}}$ | $\mathbf{83.44_{\pm 0.02}}$ | $\mathbf{61.00_{\pm 0.31}}$ |

**Model Details**: We use the Vision Transformer (ViT-B/16)(Neil & Dirk, 2020) pre-trained on the ImageNet-21K dataset (Feng et al., 2023; Yang et al., 2023) as our base model. ViT-B-16 was originally trained on images with a resolution of 224x224 pixels, utilizing a patch size of 16. To maintain compatibility, we resize our input images to 224x224 pixels. In the prompt-tuning stage, we specifically focus on optimizing shared and class prompts, as well as the classifier head. The hyper-parameter details are in Sec. A.3.2 of the appendix.

**Baselines**: We compared our method against several global and personalized Federated Learning (FL) methods that use the Prompt Tuning including Head-Tuning(Sun et al., 2022), FedPR (Feng et al., 2023),

FedVPT and FedVPT-D (Jia et al., 2022), pFedPG (Yang et al., 2023) and SGPT (Deng et al., 2024). To thoroughly assess our method's performance, we introduce a new baseline called P-PT which personalizes the prompts, giving insights into how the personalization of prompts impacts performance.

**Evaluation Methodology**: We use two key metrics to assess the performance of both the baselines and our proposed method (Deng et al., 2024): (1) *Mean Accuracy* calculates the average accuracy across individual clients' test data, reflecting adaptation to diverse client data distributions. (2) *Worst Local Accuracy* reflects the performance of the worst-performing client, indicating adaptation to the most challenging local data. Furthermore, we utilize a *heldout evaluation* strategy (Yuan et al., 2021), where 90% of clients participate in training and 10% are reserved for testing. All aforementioned metrics are reported separately for both participating and heldout clients. This setting demonstrates the model's effectiveness in onboarding new clients and adapting to previously unseen data without sharing updates with the central server. All experiments are done over 3 different runs, and the mean and standard deviations are reported as ($mean_{\pm std}$).

## 5.1 Results and Discussion

### 5.1.1 Label Heterogeneity Results

The class heterogeneity results are presented in Table- 1. PEP-FedPT outperforms all the baselines, e.g., when compared against pFedPG with CIFAR-100 pathological setting we observe an improvement of 2.5% in mean accuracy. For TinyImagenet the improvement is 8.59% (mean accuracy) over pFedPG and 17.5% (worst accuracy) over FedVPT-D. This shows that our prompt-mixing mechanism effectively addresses data heterogeneity independently and performs even better in scenarios of higher heterogeneity or lower class overlap between clients. Similar improvements are seen in other settings. Additional experiments are provided in Sec. A.4 of the appendix. The visualization of the accuracy vs communication rounds is shown in Sec. A.5.1.

Table 2: Experimental results on DomainNet and iNaturalist. For DomainNet we consider each domain belonging to a client and we report the accuracy attained by each client and the average accuracy. On the iNaturalist dataset we report the average test accuracy of all the clients and the $15^{th}$ percentile worst accuracy. Our method significantly outperforms all the baselines on this challenging dataset.

| Datasets | DomainNet(%) ↑ | | | | | | | | iNaturalist(%) ↑ | |
|---|---|---|---|---|---|---|---|---|---|---|
| Method | Clipart | Infograph | Painting | Quickdraw | Real | Sketch | Mean Acc | Worst Acc | Mean Acc | Worst Acc (15%) |
| Head-Tuning | $91.16_{\pm0.92}$ | $57.45_{\pm1.27}$ | $91.39_{\pm0.24}$ | $74.94_{\pm0.28}$ | $96.68_{\pm0.42}$ | $86.46_{\pm1.42}$ | $83.71_{\pm1.27}$ | $38.88_{\pm3.70}$ | $49.41_{\pm0.41}$ | $20.56_{\pm0.47}$ |
| FedVPT | $90.84_{\pm1.37}$ | $58.56_{\pm0.60}$ | $92.25_{\pm0.67}$ | $77.81_{\pm0.30}$ | $96.78_{\pm0.48}$ | $88.24_{\pm0.89}$ | $84.23_{\pm0.72}$ | $37.02_{\pm2.44}$ | $52.22_{\pm0.50}$ | $23.21_{\pm0.55}$ |
| FedVPT-D | $94.01_{\pm1.05}$ | $63.29_{\pm0.81}$ | $\mathbf{93.45}_{\pm0.49}$ | $84.56_{\pm1.59}$ | $96.96_{\pm0.66}$ | $91.58_{\pm0.09}$ | $87.31_{\pm0.51}$ | $42.49_{\pm1.85}$ | $57.96_{\pm1.12}$ | $30.00_{\pm0.74}$ |
| FedPR | $91.62_{\pm0.91}$ | $56.20_{\pm1.01}$ | $91.16_{\pm1.17}$ | $73.72_{\pm0.62}$ | $96.66_{\pm0.39}$ | $86.41_{\pm1.01}$ | $82.95_{\pm1.26}$ | $35.18_{\pm3.70}$ | $41.25_{\pm2.31}$ | $08.62_{\pm0.39}$ |
| SGPT | $92.64_{\pm0.65}$ | $60.62_{\pm0.22}$ | $91.54_{\pm0.78}$ | $83.55_{\pm1.85}$ | $96.55_{\pm0.17}$ | $89.93_{\pm0.47}$ | $85.56_{\pm0.60}$ | $37.34_{\pm1.41}$ | $55.78_{\pm0.57}$ | $27.27_{\pm0.32}$ |
| pFedPG | $92.89_{\pm0.82}$ | $63.56_{\pm0.88}$ | $92.27_{\pm1.01}$ | $\mathbf{87.33}_{\pm0.21}$ | $97.16_{\pm0.25}$ | $89.34_{\pm0.30}$ | $87.40_{\pm0.30}$ | $52.05_{\pm0.32}$ | $52.42_{\pm2.59}$ | $12.54_{\pm0.1}$ |
| P-PT | $90.11_{\pm1.41}$ | $56.73_{\pm1.16}$ | $90.25_{\pm1.09}$ | $74.81_{\pm0.81}$ | $95.18_{\pm0.70}$ | $85.26_{\pm0.53}$ | $82.30_{\pm1.31}$ | $35.67_{\pm1.33}$ | $45.69_{\pm0.52}$ | $16.60_{\pm0.30}$ |
| PEP-FedPT(Ours) | $\mathbf{95.46}_{\pm0.41}$ | $\mathbf{71.68}_{\pm1.41}$ | $93.00_{\pm0.55}$ | $86.89_{\pm1.53}$ | $\mathbf{97.67}_{\pm0.53}$ | $\mathbf{91.79}_{\pm0.79}$ | $\mathbf{89.15}_{\pm0.70}$ | $\mathbf{59.79}_{\pm2.52}$ | $\mathbf{63.48}_{\pm1.10}$ | $\mathbf{41.10.}_{\pm0.48}$ |

### 5.1.2 Feature Heterogeneity Results

Feature-heterogeneity results are presented in Table 2. FedVPT-D serves as strong baseline due to its inclusion of prompts at each layer. We present the results for our method. In this setting, personalized method pFedPG also proves highly beneficial due to the pronounced feature imbalance among clients. Our method PEP-FedPT improves on average by 5.52% over the best performing baseline FedVPT-D on the iNaturalist dataset. For the iNaturalist dataset, under a low client participation rate (1%), many clients are sampled infrequently, and a large number of clients (over 150) have extremely small test sets (fewer than 10 samples). Thus, misclassification of only a few test samples leads to a worst-client accuracy of zero for all methods. Consequently, the standard worst-client metric becomes uninformative. To address the evaluation difficulty in this setting, we report the client accuracy at the $15^{th}$ percentile as a redefined worst-client metric, which provides a more informative assessment of performance on underperforming clients.

### 5.1.3 Heldout Evaluation

In the Table 3 we present the results for the heldout evaluation, where we report the accuracies of the participating clients and the new clients. Participating clients are the ones who participate in the federated training and the new clients do not participate in the FL training. In this setup, the testing accurcay implies the zero-shot predictions on the held-out clients' test data. We consider the CIFAR-100 dataset

with pathological partitioning where each client observes only 10 classes. Most baseline methods achieve competitive accuracy on participating clients but show reduced performance on unseen clients. Personalized methods like pFedPG achieve very high participating accuracy but fail in held-out testing resulting in poor performance, highlighting poor generalization. Since pFedPG is not explicitly designed for this evaluation protocol, its performance under this setting should not be interpreted as indicative of its effectiveness. We include pFedPG to illustrate the behavior of personalization-based methods in a strict zero-shot setting, and therefore mark the corresponding results as Not Applicable (NA). In contrast, our method consistently achieves the best results across both datasets, with 95.66% vs. 93.71% on CIFAR-100 and 92.53% vs. 90.60% on Tiny-ImageNet, showing strong generalization to unseen clients. The results on DomainNet and iNaturalist are provided in Sec. A.4.7 of the appendix.

Table 3: Quantitative comparisons on CIFAR-100 and Tiny-ImageNet with held out evaluation: We report the accuracy with pathological partitioning where each client observes 10 classes. It can be observed that the personalized methods like pFedPG perform the worst in the held-out evaluation Our method performs well on the clients participating in the FL training and also on the unseen clients.

| Method | CIFAR-100 ($\uparrow$) | | Tiny-ImageNet ($\uparrow$) | |
|---|---|---|---|---|
| | Participating Acc | Testing Acc | Participating Acc | Testing Acc |
| Head | $77.81_{\pm 0.25}$ | $77.10_{\pm 0.41}$ | $67.97_{\pm 0.66}$ | $68.97_{\pm 0.70}$ |
| FedVPT | $83.62_{\pm 0.24}$ | $82.39_{\pm 1.12}$ | $74.15_{\pm 0.47}$ | $74.15_{\pm 1.19}$ |
| FedVPT-D | $85.06_{\pm 0.51}$ | $84.87_{\pm 0.44}$ | $77.38_{\pm 1.35}$ | $76.89_{\pm 0.90}$ |
| FedPR | $81.62_{\pm 0.27}$ | $80.61_{\pm 0.68}$ | $69.37_{\pm 1.42}$ | $68.23_{\pm 1.90}$ |
| SGPT | $83.90_{\pm 0.23}$ | $83.63_{\pm 0.64}$ | $76.38_{\pm 0.68}$ | $78.10_{\pm 1.85}$ |
| pFedPG | $93.32_{\pm 0.85}$ | $NA$ | $86.09_{\pm 1.42}$ | $NA$ |
| P-PT | $75.97_{\pm 1.38}$ | $72.19_{\pm 0.58}$ | $60.76_{\pm 0.39}$ | $60.41_{\pm 1.40}$ |
| PEP-FedPT(Ours) | $\mathbf{95.66}_{\pm 0.17}$ | $\mathbf{93.71}_{\pm 0.40}$ | $\mathbf{92.53}_{\pm 0.35}$ | $\mathbf{90.60}_{\pm 0.61}$ |

## 5.2 Analysis of PEP-FedPT

### 5.2.1 Ablations

We conducted ablation studies on shared and class-specific prompts evaluating their impact by varying the number of shared prompts and the influence of class priors on the prompt mixing strategy. Table 4 reports the effect of combining shared prompts with CCMP under both Pathological and Dirichlet splits. For the Pathological split, the average accuracy improves from 83.62% with only shared prompts to 95.46% with shared + CCMP. Similarly, for the Dirichlet split, the performance increases from 84.91% to 88.75%. These results highlight the consistent benefit of incorporating CCMP across different data partitioning strategies.

Table 4: Shared and CCMP ablation on the CIFAR-100 dataset with Dirichlet and Pathological Partitions. We report the Accuracy(%).

| Prompt Strategy | Pathological Split ($\uparrow$) | Dirichlet Split ($\uparrow$) |
|---|---|---|
| Only Shared | $83.62_{\pm 0.02}$ | $84.91_{\pm 0.07}$ |
| Shared + CCMP | $95.46_{\pm 0.16}$ | $88.75_{\pm 0.25}$ |

The Table 5 presents the effect of incorporating class priors into the prompt design on CIFAR-100 under both Pathological and Dirichlet splits. The results show that using Shared + CCMP with Class Priors (CP) consistently improves performance over the baseline without CP. In particular, the Pathological split benefits, with accuracy increasing from 84.01% to 95.46%, while the Dirichlet split also shows a notable gain from 86.12% to 88.75%. Similar analysis for other datasets is given in section A.4.3 and A.4.2 of the appendix. Further additional experiments can be found in Section A.4 of the appendix.

Table 5: Impact of Class Priors on CIFAR-100 dataset with Dirichlet and Pathological Partitions.

| Prompt Strategy | Pathological Split ($\uparrow$) | Dirichlet Split ($\uparrow$) |
|---|---|---|
| Shared + CCMP Without CP | $84.01_{\pm 0.04}$ | $86.12_{\pm 0.13}$ |
| Shared + CCMP With CP | $95.46_{\pm 0.16}$ | $88.75_{\pm 0.25}$ |

### 5.2.2 Computation and Communication

We denote that $d$ and $d_h$ are token and attention head dimensions, $C$ and $L$ denote the number of classes and layers respectively, and $T$ are the tokens. The minimum computations required by ViT forward is given as:

Table 6: Comparison of computation and communication. Resources required to achieve 83% accuracy on CIFAR-100.

| Method | Training Time (sec) ↓ | Params Communicated ↓ | Rounds Required ↓ |
|---|---|---|---|
| FedVPT | 4550 | 7.7 M | 100 |
| FedVPT-D | 4760 | 7.67 M | 90 |
| SGPT | 8170 | 13 M | 90 |
| FedPR | >5360 | >8.4 M | >100 |
| PEP-FedPT (Ours) | **1153** | **4.6 M** | **12** |

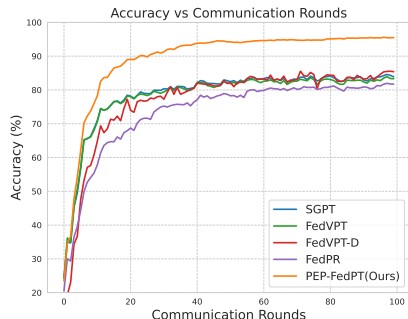

Figure 3: Convergence on CIFAR-100 with pathological partitioning.

The Query (Q), Key (K) and Value (V) requires $Tdd_h$ multiplications each. The inner product matrix $QK^T$ requires $T^2 d_h$ multiplications. The feedforward computations requirement is $d^2 T$. If $H$ heads are present and there are $C$ classes for the classification then we need $LH(3Tdd_h + T^2 d_h) + LTd^2 + Cd$ multiplications. For CIFAR-100 on ViT-B/16 the CCMP computation takes only 0.008% of total computations, which is very negligible implying the efficiency of the CCMP computation. Table 6 compares the computational and communication complexity of our proposed method, PEP-FedPT against the different baselines. For a fair comparison, we compare the methods that use global prompts. We analyze the resources required to achieve 83% accuracy, which is the highest accuracy reported for FedVPT. Our results show that PEP-FedPT achieves this accuracy in just 12 rounds, requiring lowest training time and significantly reducing communication overhead (4.6M) compared to SGPT (13.0M), where M denotes million. The claim of 12 rounds can be verified in the Figure 3. FedPR only attains 81.66% in 100 rounds so we report this as $(> 100)$. The training times reported are measured on an Nvidia RTX-A6000 GPU. The detailed computation of why **4.6** M is : Head requires $(100 \times 768)$, shared prompt $(1 \times 768)$ class prompts $(100 \times 768)$, prototypes $(100 \times 768 \times 3)$ scaled by 3 because of three layers with CCMP prompts. Total rounds 12 and in total, yields 4.6M parameters. In this communication analysis, since our method shares the global prompts, we compared only the methods that are not personalized.

## 6 Limitations and Scope for Future Work

Our approach relies on empirical estimates of class priors and `cls` token centroids, which require access to labeled data on the client side and introduce several limitations. In semi-supervised or unsupervised settings, where labeled data is scarce or unavailable, these estimates may become unreliable, potentially degrading prompt construction and making adaptation to such scenarios non-trivial. Similarly, in long-tailed data distributions, the limited presence of certain classes can negatively affect centroid quality and, consequently, model performance. Finally, the method is primarily designed for non-IID data distributions, leveraging heterogeneity to enable personalization via global parameters; under IID settings, this advantage diminishes and the method effectively reduces to standard Fed-VPT. Our theoretical result in Proposition. 1 is relatively tighter for label heterogeneity setting in comparison to feature drift owing to our Assumption.3. Exploring extensions to address these limitations presents a promising direction for future research.

## 7 Conclusion

We propose a novel prompt-tuning methodology for Vision Transformers (ViTs) by introducing class-specific prompts alongside shared prompts. Our approach leverages the `cls`-token representations in pretrained ViT layers to extract prototypes, which are then combined with each client's prior label distribution to compute soft scores that guide the mixing of class-specific prompts into a unified, optimized prompt (CCMP). This dynamic mixing allows (CCMP) to achieve personalization while using global prompts only. This combined prompt (CCMP) is subsequently embedded within the ViT layer. Our method PEP-FedPT, achieves State of the Art performance, surpassing previous methods across the benchmark datasets.[4]

[4]**Acknowledgment:** This work is partially supported by the P3DX project, seed-funded by the Ministry of Electronics and Information Technology (MeitY). The authors would also like to acknowledge compute supports received from Kotak-IISc AI-ML Centre (KIAC), IISc and the PMRF fellowship.

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

# A  Appendix

## A.1  Overview of Notation and Definitions

We give the overview of the notations and definitions used in the paper

- Boldface letters to denote matrices and vectors.

- "·" refers to element-wise multiplication.

- $*$ refers to the standard matrix multiplication.

- We define $TL_i$ as the $i^{th}$ transformer layer.

- $\mathbb{E}$ denotes the expectation operator.

- The term $\delta_k^c$ represents the prior probability of class $c$ occurring at client $k$.

- Additionally, $sim(\mathbf{p}, \mathbf{q})$ denotes the cosine similarity, while $\mathbb{I}$ represents the indicator function.

- $\boldsymbol{\mu}_{l-1,k,t}^c$ denotes the client $k$ computing the CLS token prototypes in communication round $t$ and the input of layer $l$.

- $\mathbf{H}$ denotes the classification head.

- $\|.\|$ denotes Euclidean norm or 2-norm.

- $[M]$ denotes the set $\{1, 2, ., .M\}$.

- $\Lambda = \{rR, rR + 1, \ldots, (r + 1)R - 1\}$ denotes the set of communication rounds in $r$-th update period of length $R$.

- $n_{k,c}$ denotes the number of samples corresponding to class $c$ in client $k$.

- $\mathbf{P_S}$ and $\mathbf{P_C}$ denote shared prompts and class-specific prompts respectively.

- $\mathbf{m}$ denotes the CCMP.

- $N_k$ denotes the total number of datapoints at client $k$ and $n$ denotes the number of clients.

## A.2 Method Details: Algorithm

We briefly go over the prototype update and the CCMP computation equations. Client level prototype aggregation at communication round $t$ is given in the below Eq. 18

$$\boldsymbol{\mu}^c_{l-1,k,t} = \begin{cases} \frac{1}{n_{k,c}} \sum_{i=1}^{N_k} \mathbf{cls}_{l-1,i,k,t} \cdot \mathbb{I}_{y_{i,k}=c}, & n_{k,c} > 0, \\ \mathbf{0}, & n_{k,c} = 0. \end{cases} \tag{18}$$

Server prototype aggregation during the warm-up phase is given by Eq. 19

$$\boldsymbol{\mu}^c_{l-1,0} = \frac{1}{|S_0|} \sum_{k \in S_0} \boldsymbol{\mu}^c_{l-1,k,0} \tag{19}$$

Server aggregation of the prototypes at the end of $r$-th update period is in Eq. 20

$$\hat{\boldsymbol{\mu}}^c_{l-1,r} = \begin{cases} \frac{1}{D_c} \sum_{t \in \Lambda} \sum_{k \in S_t} \mathbb{I}_{\{\boldsymbol{\mu}^c_{l-1,k,t} \neq \mathbf{0}\}} \boldsymbol{\mu}^c_{l-1,k,t}, & D_c > 0, \\ \mathbf{0}, & D_c = 0. \end{cases} \tag{20}$$

Sever updating the prototypes based on the momentum is given in Eq. 21. If $D_c$ is 0, we set $\rho = 1$.

$$\boldsymbol{\mu}^c_{l-1,r} = \rho \cdot \boldsymbol{\mu}^c_{l-1,r-1} + (1 - \rho) \cdot \hat{\boldsymbol{\mu}}^c_{l-1,r} \tag{21}$$

The class prior for class $c$ at client $k$ is computed as in Eq. 22

$$\delta^c_k = \frac{1}{N_k} \sum_{i=1}^{N_k} \mathbb{I}_{y_{i,k}=c} \tag{22}$$

The soft scores are computed based on similarity between the class prototypes and cls representations as in Eq. 23

$$\hat{s}^c_{i,l-1,k} = exp\left(\frac{sim\left(\mathbf{cls}_{l-1,i,k}, \boldsymbol{\mu}^c_{l-1}\right)}{\tau}\right) \delta^c_k \tag{23}$$

The scores are converted to probabilities using Eq. 24

$$s_{i,l-1,k}^c = \frac{\hat{s}_{i,l-1,k}^c}{\sum_{j=1}^{|C|} \hat{s}_{i,l-1,k}^j} \tag{24}$$

The probabilities serve as weights of class-specific prompts which produce the Class Contexualized Mixed Prompts (CCMP) as in Eq. 25

$$\mathbf{m}_{l-1} = \mathbf{P}_C * \mathbf{s}_{i,l-1,k} \tag{25}$$

$\mathbf{s}_{i,l-1,k}$ is the vector containing $s_{i,l-1,k}^c$ for different values of $c$.

---

**Algorithm 2:** PEP-FedPT

---

**Input:** $\mathbf{H}$, $\mathbf{P_S}$,$\mathbf{P_C}$ $\mu$, Pretrained Vision Transformer $\mathbf{w}_{pre}$,Training data $(x, y) \sim \mathcal{D}$, Set of class labels $C$, Learning rate $\eta$, Number of local epochs $E$, Update period $R$, Total number of communication rounds $T$, Total number of clients $n$, CCMP layer index $l$

**Output:** $\boldsymbol{\theta} = \{\mathbf{H}, \mathbf{P_S}, \mathbf{P_C}\}, \boldsymbol{\mu}_{l-1}$

**1** Server samples the subset $S_0 \subset [n]$
**2** $\boldsymbol{\mu}_{l-1}^c \leftarrow WarmStartUp(\mathbf{w}_{pre}, S_0) \ \forall c \in C$
**3 for** *round* $t \in [T]$ **do**
**4**     Server samples participating clients $S_t \subset [n]$
**5**     **for** *client* $k \in [S_t]$ **do**
**6**        $\boldsymbol{\theta}_k^t, \boldsymbol{\mu}_{l-1,k,t}^c = \texttt{LocalTrain}(\boldsymbol{\theta}^t, \boldsymbol{\mu}_{l-1}, l, k, t); \ \forall c \in C$

**7** $\boldsymbol{\theta}^{t+1} \leftarrow FedAveraging(\{\boldsymbol{\theta}_k^t, k \in S_t\})$McMahan et al. (2017)

**8 if** $t \mod R = 0$ **then**
**9**     $r \leftarrow \frac{t}{R}$
**10**    **for** $c \in C$ **do**
**11**       $\hat{\boldsymbol{\mu}}_{l-1,r}^c \leftarrow AggregateCentroids(\{\boldsymbol{\mu}_{l-1,k,t}^c, t \in \Lambda\})$ [Eq. 20]
**12**       $\boldsymbol{\mu}_{l-1,r}^c \leftarrow UpdateCentroids(\hat{\boldsymbol{\mu}}_{l-1,r}^c, \boldsymbol{\mu}_{l-1,r-1}^c);$ [Eq.21]
**13**       $\boldsymbol{\mu}_{l-1}^c \leftarrow \boldsymbol{\mu}_{l-1,r}^c$

**14** Return $\boldsymbol{\theta}, \boldsymbol{\mu}_{l-1}$
**15 Function** $\texttt{WarmStartUp}(\mathbf{w}_{pre}, S_{in}, l)$**:**
**16**    **for** *client* $k$ *in* $S_{in}$ **do**
**17**      obtain $\boldsymbol{\mu}_{l-1,k,0}^c$ [Eq. 10]
**18**      Return $\boldsymbol{\mu}_{l-1,k,0}^c \forall c \in C$
**19**    Server obtains $\boldsymbol{\mu}_{l-1,0}^c$ [Eq. 19]
**20**    Return $\boldsymbol{\mu}_{l-1,0}^c$

**21 Function** $\texttt{LocalTrain}(\boldsymbol{\theta}^t, \boldsymbol{\mu}_{l-1}, l, k, t)$**:**
**22**    compute $\boldsymbol{\mu}_{l-1,k,t}^c \ \forall c \in C$ Eq. 10
**23**    $\boldsymbol{\theta}_k \leftarrow \boldsymbol{\theta}^t$
**24**    **for** $e = 1 \rightarrow E$ **do**
**25**      $\mathbf{m} \leftarrow PromptMixing((\mathbf{H}, \mathbf{P_S}, \mathbf{P_C}, \mathbf{w}_{pre}, \boldsymbol{\mu}_{l-1})$ [Eq. 22, 23, 24, 25]
**26**      Define loss $l = l(\mathbf{H}, \mathbf{P_S}, \mathbf{m}, x, y)$
**27**      $\boldsymbol{\theta}_k \leftarrow \boldsymbol{\theta}_k - \eta \cdot \nabla l_{\boldsymbol{\theta}_k}$
**28**    Return $\boldsymbol{\theta}_k, \boldsymbol{\mu}_{l-1,k,t}^c$

---

### A.3  Experimental Setup

#### A.3.1  Details on Heterogeneity

We consider two different kinds of heterogeneity label imbalance and feature imbalance. In the label imbalance we again consider two different settings, pathological and the Dirichlet based non-iid settings as shown in Figure 4.

For pathological settings we select few classes of data points for each client and allocate the data among those labels. For Dirichlet we allocate the data by drawing a sample from the Dirichlet distribution. We consider these settings using the CIFAR-100 and Tiny-ImageNet Datasets by distributing the data among the 100 and 200 clients respectively and sampling only 5 clients in each communication round. For Dirichlert settings the degree of non-iid is controlled by the parameter $\delta$ and its denoted by $Dir(\delta)$. The lower delta implies higher heterogeneity and higher value implies the lower heterogeneity. Throughout the work we consider the value of $\delta$ to be 0.3.

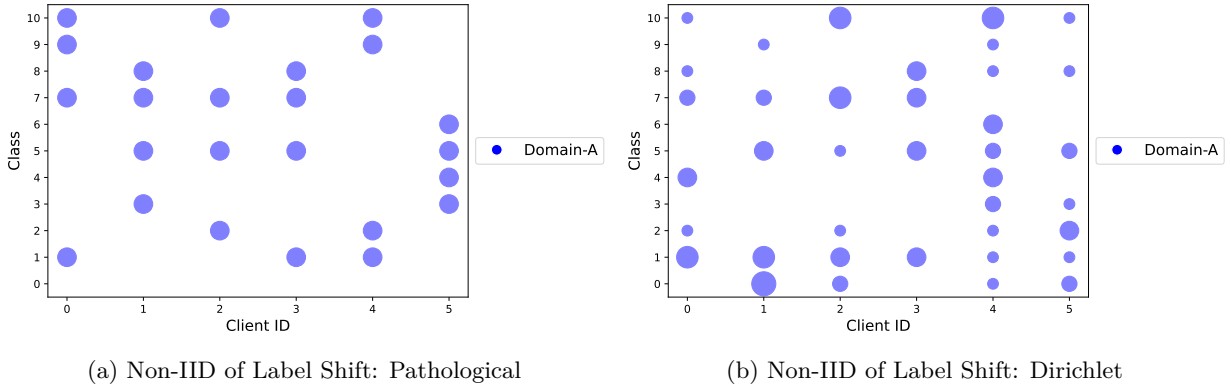

(a) Non-IID of Label Shift: Pathological        (b) Non-IID of Label Shift: Dirichlet

Figure 4: Comparison of Non-IID Label Shift due to Pathological setting and the Dirichlet setting

By feature imbalance, we mean clients are distributed with different domains. It can be seen in the Figure. 5. The DomainNet dataset can be viewed as analogous to the one described in the Figure 5a. In the Figure 5b the split shows the mix of feature and the label imbalance.

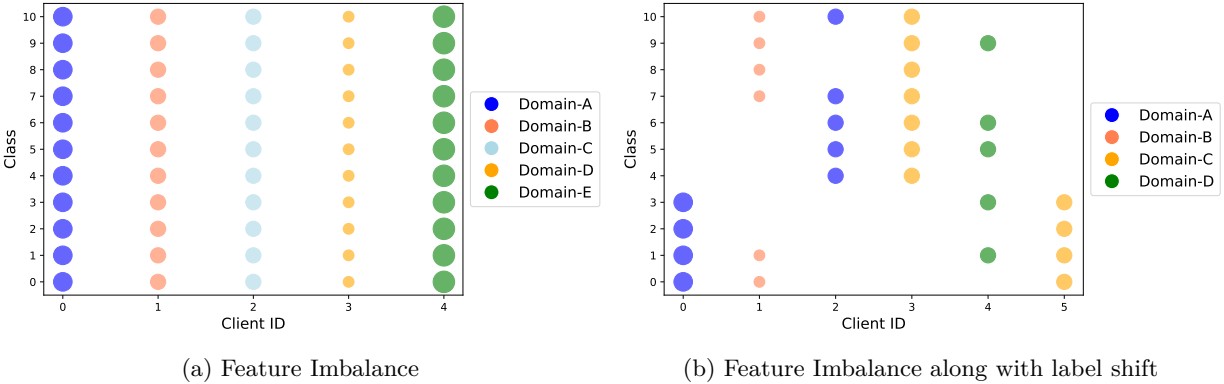

(a) Feature Imbalance        (b) Feature Imbalance along with label shift

Figure 5: Comparison of Non-IID Feature Shift

#### A.3.2  HyperParameter Details

We follow stochastic Gradient Descent with momentum (Deng et al., 2024) as the default optimizer with learning rate 0.1 with exponential decay and the momentum 0.9. For all the experiments we consider number of shared prompts ($n_S$) to be 1, unless explicitly mentioned. We add the class specific prompts at the layers

5, 6 and 7. We also set the gradient clipping to 10 following Acar et al.. For all our experiment we consider number of shared prompts to 1 except the Tiny-ImageNet Dirichlet where we set it to 5. The CCMP is inserted at the layers 5, 6 and 7. We set the the temperature parameter $\tau$ to 0.05 for all our experiments. We show the dataset-specific hyperparameters in the table.

Table 7: Dataset-specific hyperparameter settings

| Dataset | # Classes | # Clients | Classes / Client | Comm. Rounds | Participation Rate | Local Epochs | Centroid Update Interval |
|---|---|---|---|---|---|---|---|
| CIFAR-100 | 100 | 100 | 10 (Pathological / Dir(0.3)) | 100 | 5% | 5 | 10 |
| Tiny-ImageNet | 200 | 200 | 10 (Pathological / Dir(0.3)) | 100 | 2.5% | 5 | 10 |
| DomainNet | 10 | 60 | 5 | 50 | 10% | 5 | 10 |
| iNaturalist | 1203 | 1018 | 10 | 500 | 1% | 2 | 10 |

### A.4 Additional Experiments

### A.4.1 Class-Level Differential Privacy via Laplace Mechanism

Table 8: Impact of class-level DP noise ($\epsilon = 0.2$) on mean accuracy across datasets.

| Method | CIFAR-100 (Path) | CIFAR-100 (Dir-0.3) | Tiny-ImageNet (Path) | Tiny-ImageNet (Dir-0.3) |
|---|---|---|---|---|
| With DP Noise | $93.23_{\pm 0.07}$ | $86.92_{\pm 0.07}$ | $91.16_{\pm 0.13}$ | $82.92_{\pm 0.11}$ |
| Without DP Noise | $95.46_{\pm 0.16}$ | $88.75_{\pm 0.25}$ | $91.52_{\pm 0.11}$ | $83.44_{\pm 0.02}$ |

To prevent leakage of individual client privacy in a federated learning setting through sharing of cls-tokens, we employ the most common Laplace mechanism as described in Dwork et al. (2006) for class prototype everytime the client shares it with the server. After a client aggregates cls-tokens and forms its corresponding class prototypes, we estimate the sensitivity of each class $c$ for a client $k$ based on the maximum L1 deviation of its CLS- token representation from the corresponding class prototype, normalized by the number of samples $N_{c,k}$:

$$S_{c,k} = \frac{2 \cdot \max_i \|\mathbf{cls}_{i,k}^{(c)} - \boldsymbol{\mu}_c\|_1}{N_{c,k}}$$

where $\mathbf{cls}_{i,k}^{(c)}$ denotes the CLS token of the $i$-th sample belonging to class $c$ at client $k$, $N_{c,k}$ is the total number of samples belonging to class c for client k and $\boldsymbol{\mu}_c$ is the prototype representing class $c$ in the embedding space across all clients. To enforce differential privacy, Laplace noise is added to each class prototype $\theta_{c,k}$ based on its sensitivity $S_{c,k}$ and a predefined privacy budget $\epsilon$:

$$\theta_{c,k} \leftarrow \theta_c + \text{Laplace}(0, S_{c,k}/\epsilon)$$

This class-aware noise injection is performed during every client-server communication, for all the CCMP layers. As a result, individual class-level contributions are obfuscated, thereby enhancing privacy while preserving model performance under non-IID data distributions. In the Table 8 we have shown the impact of dp noise on our overall accuracy. We have used $\epsilon = 0.2$ for our experiment.

**Theoretical guarantees for differential privacy**: The sensitivity function defined in A.4.1 is an upper bound of the true maximum possible difference of average calculated from neighboring datsets.

Consider two datasets $\mathcal{D}_1$ and $\mathcal{D}_2$, each containing an equal number $N$ of `[cls]` token representations, that differ at exactly one data point: $\mathbf{cls}_1 \in \mathcal{D}_1$ and $\mathbf{cls}_2 \in \mathcal{D}_2$. The sensitivity of the dataset average is given by

$$\Delta f = \frac{\max \|\mathbf{cls}_1 - \mathbf{cls}_2\|_1}{N}.$$

For any $\boldsymbol{\mu}$

$$\Delta f = \frac{max \|\mathbf{cls_1} - \boldsymbol{\mu} + \boldsymbol{\mu} - \mathbf{cls_2}\|_1}{N} \leq \frac{max \|\mathbf{cls_1} - \boldsymbol{\mu}\|_1 + \|\boldsymbol{\mu} - \mathbf{cls_2}\|_1}{N} \leq \frac{2 \cdot \max_i \|\mathbf{cls}_i - \boldsymbol{\mu}\|_1}{N}$$

By taking $\boldsymbol{\mu}$ to be the empirical average of all cls-token representations. We upper bound the sensitivity by $S_{c,k}$. Following this, it is straightforward to establish privacy guaranties. We refer to Dwork & Roth (2014) for the formal proof. In particular, Theorem 3.6 shows that adding Laplace noise to the class prototype ensures $(\epsilon, 0)$-differential privacy.

### A.4.2   Impact of Class Priors

Table 9: Ablation on class priors for iNaturalist, DomainNet and Tiny-ImageNet datasets. We report the Mean Accuracy (%)

| Prompt | iNaturalist | DomainNet | Tiny-ImageNet |
|---|---|---|---|
| Shared + CCMP Without CP | $54.38_{\pm 0.56}$ | $86.34_{\pm 0.52}$ | $81.08_{\pm 0.13}$ |
| Shared + CCMP With CP | $63.48_{\pm 1.10}$ | $89.15_{\pm 0.70}$ | $83.44_{\pm 0.02}$ |

The ablation results in the Table 9 highlight the effect of incorporating class priors into the Shared+CCMP strategy. For both iNaturalist and DomainNet, adding class priors consistently improves mean accuracy compared to using Shared+CCMP without priors, with gains of nearly 9% on iNaturalist and about 3% on DomainNet.

### A.4.3   Impact of CCMP and Shared Prompts

Table 10: Ablation on prompts for iNaturalist, DomainNet and Tiny-ImageNet datasets.

| Prompt | iNaturalist | DomainNet | Tiny-ImageNet |
|---|---|---|---|
| Only Shared | $52.22_{\pm 0.50}$ | $84.23_{\pm 0.72}$ | $79.02_{\pm 0.34}$ |
| Shared + CCMP | $63.48_{\pm 1.10}$ | $89.15_{\pm 0.70}$ | $83.44_{\pm 0.02}$ |

The ablation results in the Table 10 compare the effect of using only shared prompts versus combining them with CCMP. On iNaturalist, the mean accuracy improves from 52.22% to 63.48%, while on DomainNet, the performance rises from 84.23% to 89.15% when CCMP is added.

### A.4.4   Impact of increasing shared prompts

Table 11: Impact of Accuracy on increasing the number of shared prompts with non-iid partitioning of $Dir(0.3)$. Increasing $n_S$ results in minor improvements for CIFAR-100 and DomainNet

| Dataset | $n_S = 1$ | $n_S = 5$ | $n_S = 10$ |
|---|---|---|---|
| CIFAR-100 | $88.75_{\pm 0.25}$ | $89.65_{\pm 0.15}$ | $90.53_{\pm 0.59}$ |
| DomainNet | $89.15_{\pm 0.70}$ | $89.29_{\pm 0.66}$ | $90.22_{\pm 0.33}$ |

In the Table 11, we show the impact of varying the number of shared prompts. It can be observed that the impact is quite minimal.

### A.4.5   On the Gain of CCMP

Introducing class-specific prompts increases the total parameter space compared to using a single global prompt. However, the performance gain achieved by our method is not solely due to this increased parameter count. To validate this, we augment the FedVPT baseline by adding 50 and 100 prompts (matching the scale of our class prompts). The mean accuracy improves initially but quickly saturates, with only marginal gains between 50 and 100 prompts. We can see this in Table 12, we have used the CIFAR-100 dataset with pathological data partitioning for this experiment. This indicates that merely increasing the number of prompt tokens is not sufficient to achieve better performance. Instead, our method's distinct soft mixing of class-specific prompts using global class prototypes and local client priors plays a key role in boosting accuracy, demonstrating the effectiveness of our proposed personalized prompt tuning mechanism.

Table 12: Effect of increasing number of prompts in FedVPT baseline. The accuracy saturates despite increasing parameter space, indicating that gains from our method are not due to higher parameter count.

| Number of Prompts | Mean Accuracy (%) |
|---|---|
| 1 | $83.62_{\pm0.02}$ |
| 50 | $87.15_{\pm0.14}$ |
| 100 | $87.45_{\pm0.11}$ |

### A.4.6 Varying the Location of CCMP Injection

In the Table 13. We perform the analysis of our method PEP-FedPT. It can be seen that adding the CCMP prompts too early in the ViT is not beneficial as the `cls` token representations at the very early layers do not have better representations. Adding the prompts at later layers is also not beneficial, even tough the `cls` tokens have better representations, since the prompts inserted are not deep enough to learn useful representations. The choice of using the three prompts is to be efficient and, at the same time, to provide a fair comparison with methods like SGPT (Deng et al., 2024).

Table 13: Impact of adding the proposed CCMP prompts at different layers of ViT on CIFAR-100.

| Position of CCMP | Mean Accuracy |
|---|---|
| 1, 2, 3 | $90.05_{\pm0.21}$ |
| 5, 6, 7 | $95.46_{\pm0.16}$ |
| 8, 9, 10 | $93.55_{\pm0.14}$ |

### A.4.7 Heldout Evaluation on DomainNet and iNaturalist

The comparison shows that methods like Fed-VPT-D and SGPT provide competitive results, especially on DomainNet. However, our method achieves the best overall performance, with 62.41% participating and 54.16% testing accuracy on iNaturalist, and 90.32% participating and 88.73% testing accuracy on DomainNet. This highlights its robustness across both datasets and evaluation settings. For iNaturalist about 916 clients participated in the training while 102 clients were held out. For DomainNet, 6 clients, one per domain, were held out, and 54 clients, 9 from each domain, participated in the training.

Table 14: Comparison of methods on iNaturalist and DomainNet datasets with the held-out setting

| Method | iNaturalist (↑) | | DomainNet (↑) | |
|---|---|---|---|---|
| | Participating Acc | Testing Acc | Participating Acc | Testing Acc |
| Head | $48.87_{\pm0.41}$ | $45.27_{\pm0.51}$ | $82.34_{\pm1.81}$ | $83.19_{\pm2.02}$ |
| Fed-VPT | $51.69_{\pm0.41}$ | $48.05_{\pm0.12}$ | $82.92_{\pm1.33}$ | $83.68_{\pm1.46}$ |
| Fed-VPT-D | $57.13_{\pm1.12}$ | $53.20_{\pm1.18}$ | $87.08_{\pm1.24}$ | $87.54_{\pm1.52}$ |
| P-PT | $43.87_{\pm0.94}$ | $41.20_{\pm1.40}$ | $82.89_{\pm0.28}$ | $83.05_{\pm2.15}$ |
| FedPR | $38.62_{\pm0.16}$ | $36.03_{\pm0.15}$ | $83.59_{\pm0.17}$ | $82.62_{\pm1.79}$ |
| SGPT | $55.82_{\pm0.12}$ | $53.81_{\pm0.12}$ | $86.55_{\pm0.58}$ | $87.27_{\pm0.69}$ |
| pFedPG | $55.61_{\pm0.12}$ | $NA$ | $88.34_{\pm0.05}$ | $NA$ |
| PEP-FedPT(Ours) | $\mathbf{62.41}_{\pm0.15}$ | $\mathbf{54.16}_{\pm0.39}$ | $\mathbf{90.32}_{\pm0.18}$ | $\mathbf{88.73}_{\pm0.63}$ |

### A.4.8 Alternative view of Worst Client Accuracy

Table. 15 reports worst-client accuracy on iNaturalist, measured as the lower-tail (5%, 10%, and 15%) percentiles of per-client test accuracy, which reflects the performance of the most disadvantaged clients under data heterogeneity. Most baseline methods achieve near-zero accuracy at the 5% percentile, indicating limited robustness. In contrast, our method consistently attains the highest worst-client accuracy across all percentiles, with substantial margins over competing approaches. The gains at the 5% percentile demonstrate a clear improvement for the worst-performing clients, while the consistent advantages at 10% and 15%

Table 15: $k\%$ percentile accuracy on iNaturalist.

| Method | 5% | 10% | 15% |
|---|---|---|---|
| Head | 0 | $12.50_{\pm0.32}$ | $20.56_{\pm0.47}$ |
| VPT | 0 | $14.28_{\pm0.16}$ | $23.21_{\pm0.55}$ |
| VPT-D | $10.50_{\pm0.20}$ | $20.00_{\pm0.23}$ | $30.00_{\pm0.74}$ |
| P-PT | 0 | $10.93_{\pm0.25}$ | $16.66_{\pm0.30}$ |
| SGPT | $05.50_{\pm0.37}$ | $18.91_{\pm0.43}$ | $27.27_{\pm0.32}$ |
| FedPR | 0 | $03.54_{\pm0.14}$ | $08.62_{\pm0.39}$ |
| pFedPG | 0 | 0 | $12.54_{\pm0.41}$ |
| PEP-FedPT(Ours) | $\mathbf{20.00}_{\pm0.38}$ | $\mathbf{33.00}_{\pm0.50}$ | $\mathbf{41.10}_{\pm0.48}$ |

percentiles indicate more equitable performance across the federation. Overall, the results highlight that the proposed approach improves robustness to client heterogeneity beyond average accuracy gains.

## A.5 Visualization and Further Analysis

### A.5.1 Accuracy Vs Communication Rounds

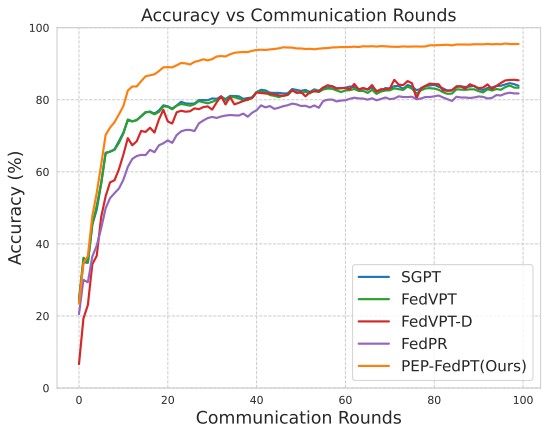

Figure 6: Comparison of the convergence of different methods across the Communication rounds on the Tiny-ImageNet dataset with pathological non-iid partitioning where each client only observes 10 classes.

The Figure 6 shows how the accuracy is improving across the FL communication rounds across the various algorithms. It is clearly evident that our proposed PEP-FedPT algorithm attains the best accuracy in fewer communication rounds compared to the other algorithms, thus minimizing the computation and communication costs.

### A.5.2 t-SNE visualization of class-prompts

In the Figure 7, we show the t-sne visualization of the trained class prompts on CIFAR-100 pathological 10-class setting and we observe that each class prompt learns its own representation, which is beneficial to making the final classification decision.

### A.5.3 Visualization of the soft weights for CCMP

In the Figure 8 we plot the soft weights averaged across all the test examples belonging to class 0 and class 1 across all the clients. It can be observed that on an average the soft scores gives high score for the relevant class prompts.

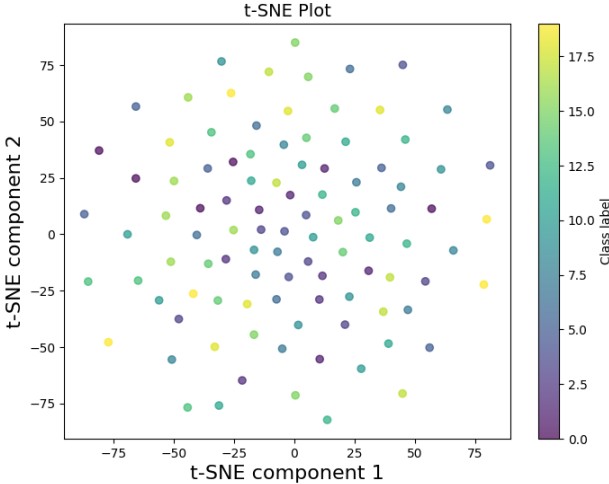

Figure 7: t-SNE visualization of the learned class prompts, it can be seen that each prompt learns its own representation implying no collapse of dimensions.

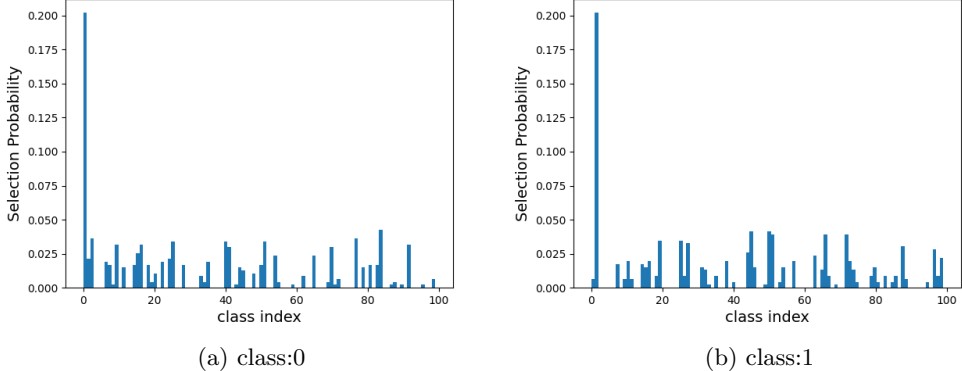

(a) class:0                    (b) class:1

Figure 8: soft weights Averaged over all the data points that belong to class 0 and 1. It shows that on Average the soft weights give more importance to the prompt corresponding to the true class.

### A.5.4 Visualization of Representations at different layers

In the Figures 9d and 9h, we observe that in initial layers the representations are uniformly distributed across the manifold post-training, which suggests that the utility of shared prompts is distinct from that of CCMP.

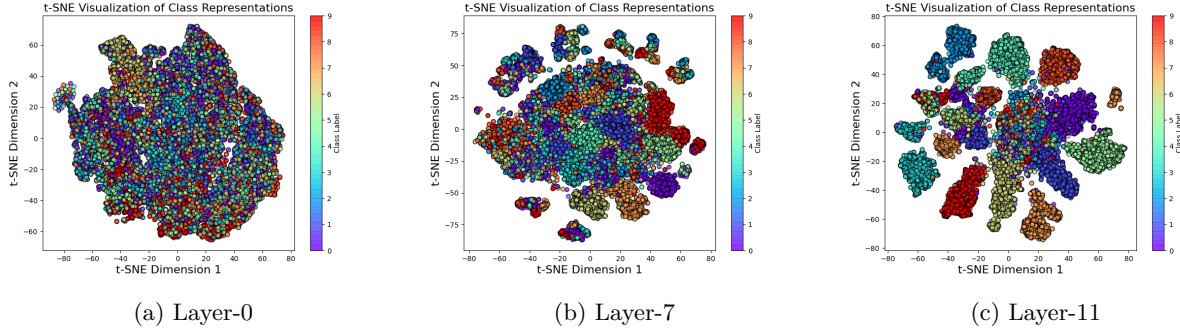

(a) Layer-0        (b) Layer-7        (c) Layer-11

(d) t-SNE representations of `cls` tokens for different layers using DomainNet dataset. It indicates that the initial layer representations are distributed uniformly over the manifold. The representation gets better once CCMP is incorporated in the later layers.

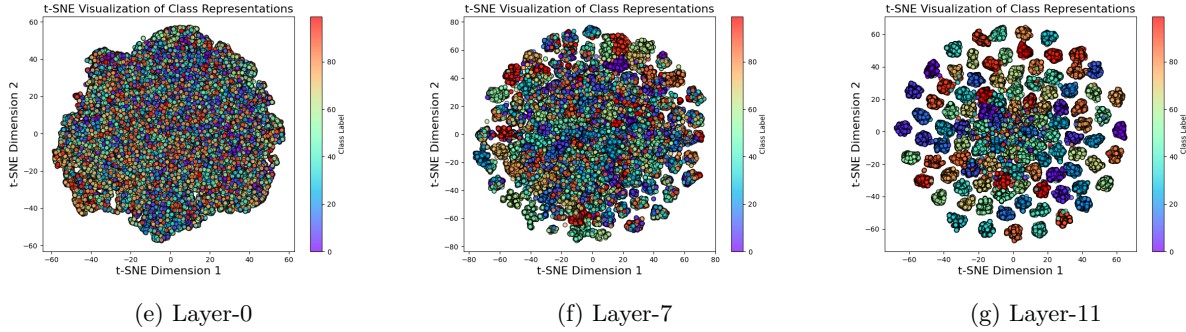

(e) Layer-0        (f) Layer-7        (g) Layer-11

(h) t-SNE representations of `cls` tokens for different layers using CIFAR-100 dataset, denoting that shared prompts at layer 0 learn only common class representations, unlike CCMP introduced later in the model. The representations are obtained using the fine-tuned ViT-B/16.

Figure 9: Comparison of t-SNE representations across layers for DomainNet and CIFAR-100 datasets.

### A.5.5 Robustness on varying the Dirichlet Concentration

Table 16: CIFAR results under different Dirichlet partitions.

| Method | Dir(0.1) | | Dir(0.5) | |
|---|---|---|---|---|
| | Mean Acc | Worst Acc | Mean Acc | Worst Acc |
| Head | $79.21_{\pm 0.12}$ | $65.48_{\pm 0.02}$ | $79.95_{\pm 0.14}$ | $72.24_{\pm 0.45}$ |
| VPT | $84.13_{\pm 0.03}$ | $71.80_{\pm 0.01}$ | $84.97_{\pm 0.01}$ | $75.63_{\pm 0.01}$ |
| VPT-D | $87.08_{\pm 0.02}$ | $74.00_{\pm 0.02}$ | $\mathbf{87.92}_{\pm 0.74}$ | $79.00_{\pm 0.12}$ |
| P-PT | $79.16_{\pm 0.24}$ | $67.24_{\pm 1.24}$ | $78.80_{\pm 0.38}$ | $64.54_{\pm 0.68}$ |
| SGPT | $85.36_{\pm 0.01}$ | $73.00_{\pm 0.16}$ | $86.04_{\pm 0.02}$ | $73.49_{\pm 0.01}$ |
| FedPR | $81.64_{\pm 0.32}$ | $65.08_{\pm 1.80}$ | $82.11_{\pm 0.59}$ | $71.42_{\pm 0.84}$ |
| pFedPG | $84.14_{\pm 0.49}$ | $73.62_{\pm 0.78}$ | $73.90_{\pm 0.38}$ | $60.00_{\pm 0.98}$ |
| PEP-FedPT(Ours) | $\mathbf{90.85}_{\pm 0.13}$ | $\mathbf{84.28}_{\pm 1.15}$ | $87.75_{\pm 0.04}$ | $\mathbf{79.51}_{\pm 0.16}$ |

This table presents the performance of different methods on the CIFAR dataset under two levels of label heterogeneity, modeled by Dirichlet partitions ($Dir(0.1)$ and $Dir(0.5)$ ). For each method, we report the average (avg) and worst-case (worst) accuracy across clients, along with the standard deviation. Our proposed method consistently achieves the highest or comparable average and best Worst Acc accuracy, indicating superior robustness and effectiveness compared to baseline methods.It can be seen that performance gain is higher when data heterogeneity is more. Our method becomes FedVPT in the iid setup as the scores used to mix class prompts will become identical.

### A.5.6 Sensitivity to Temperature $\tau$

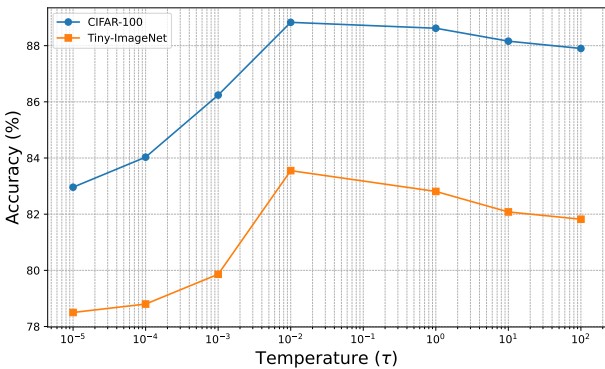

Figure 10: Sensitivity of the Average Accuracy to the $\tau$

Figure. 10 shows the effect of the temperature parameter, $\tau$, on model accuracy for two datasets: CIFAR-100 and Tiny-ImageNet. Accuracy is measured for different values of $\tau$ ranging from $10^{-5}$ to 100. For both datasets, increasing $\tau$ initially improves accuracy, reaching a peak at the same $\tau$), after which further increases in $\tau$ lead to a drop in performance. This indicates that a moderate temperature helps optimize model performance, while very small or very large temperatures can reduce accuracy.

### A.5.7 Impact of temperature on optimal location for CCMP injection

Table 17: Impact of temperature $\tau$ on optimal location for CCMP injection: Accuracies shown in CIFAR-$Dir(0.3)$ setting

| Layers | $\tau = 0.0001$ | $\tau = 0.05$ | $\tau = 100$ |
|---|---|---|---|
| 1, 2, 3 | $86.18_{\pm 0.05}$ | $86.70_{\pm 0.04}$ | $86.32_{\pm 0.16}$ |
| 5, 6, 7 | $86.24_{\pm 0.14}$ | $88.75_{\pm 0.25}$ | $87.90_{\pm 0.02}$ |
| 9, 10, 11 | $87.77_{\pm 0.04}$ | $85.68_{\pm 0.34}$ | $87.35_{\pm 0.18}$ |

In table 17 we show how the accuracy varies on CIFAR-100 under dirichlet setting. At a low temperature setting ($\tau = 0.0001$), the similarity scores computed from the `cls` token representations receive a significantly higher relative weight. In this regime, performance is primarily driven by the quality of these scores rather than the depth at which CCMP is injected. Since `cls` token representations at later layers (e.g., layers 9–11) are more expressive, they provide more accurate similarity estimates, resulting in improved accuracy. As the temperature increases ($\tau = 0.05$ and $\tau = 100$), the influence of the similarity scores is reduced, and the depth of prompt insertion becomes a critical factor. At higher temperatures, sufficient insertion depth is required to enable the prompts to learn meaningful representations, and shallow insertion is no longer adequate to achieve strong performance.

### A.5.8 Personaization and generalization Trade-off

Figure. 11 visualizes the tradeoff between personalized performance and generalized performance across four datasets: CIFAR-100, Tiny-ImageNet, iNaturalist, and DomainNet. Each point corresponds to a method, with

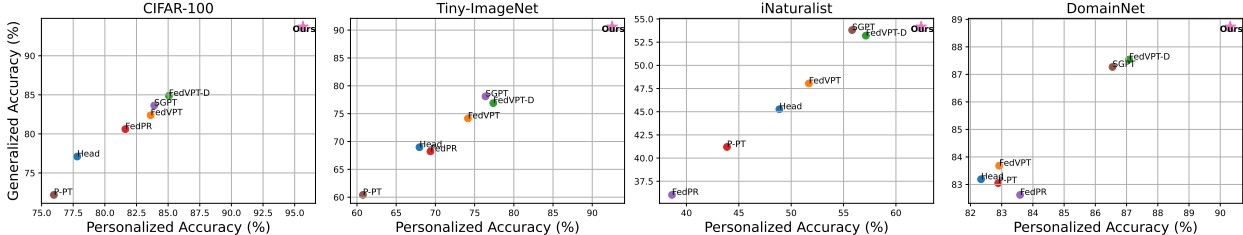

Figure 11: Personalization and Generalization Trade-off of different methods

the x-axis representing personalized accuracy (participating accuracy on clients) and the y-axis representing generalized accuracy (testing accuracy on held-out data). Methods closer to the top-right corner achieve a better balance between personalization and generalization. Across all datasets, PEP-FedPT consistently occupies dominant region, achieving simultaneously higher personalized and generalized accuracy compared to prior methods. In contrast, several baselines improve personalization at the expense of generalization or vice versa, highlighting an inherent tension between the two objectives. The results demonstrate that PEP-FedPT bridges this tradeoff.

### A.5.9 Evolution of Class Prompts across rounds

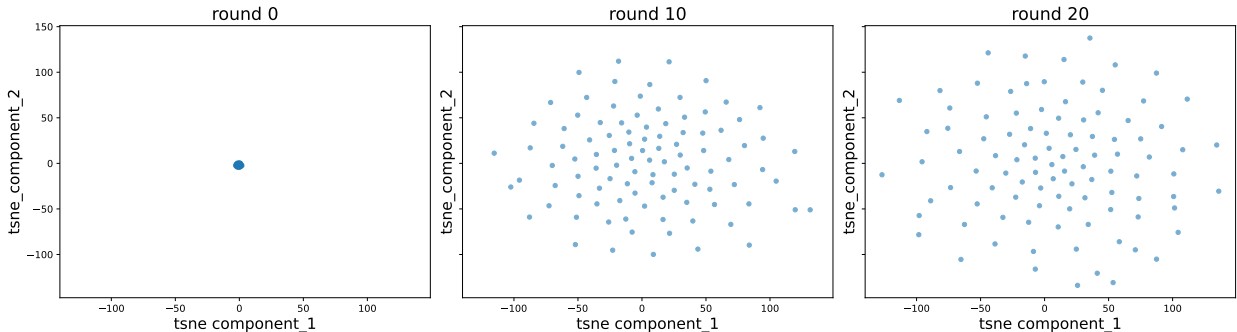

Figure 12: Evolution of Class Prompts across Rounds on CIFAR-100 Dataset

The Figure. 12 illustrates the evolution of class prompts across training rounds on the CIFAR-100 dataset. At the initial stage (round 0), all class prompts are tightly clustered, indicating that they start from a nearly identical or uninformative initialization. As training progresses (round 10), the prompts gradually spread out, reflecting the model's ability to differentiate between classes. By later rounds (round 20), the prompts form well-separated representations, suggesting that each class prompt has adapted to capture class-specific semantic information.

## A.6 Theoretical Details

### A.6.1 CCMP as minimizer of quadratic upper bound around class prompts

For clarity and completeness, we restate the relevant proposition from the main paper.

**Proposition 3.** *If the assumptions 1 to 3 hold, we show that $f$ can be upper bounded as $f \leq \tilde{L} = \frac{1}{n}\sum_{k=1,i=1}^{n,|C|} \delta_k^i \left( l_k^i(\mathbf{p}_{c_i}) + \frac{\beta_{\max}}{2} \|\mathbf{m}(k) - \mathbf{p}_{c_i}\|^2 \right) + C$ and it is minimized at $\mathbf{m}(k) = \sum_{i=1}^{|C|} \delta_k^i \mathbf{p}_{c_i}, \quad \forall k \in [n]$. which is equivalent to the (CCMP) described in sec.4.2 as $\tau >> 1$. $\beta_{\max} = \max_{i \in [|C|]} \beta_i$, $C$ is a constant which depends on $\mathcal{P}$. This vanishes when $\mathbf{p}_{c_i} = \mathbf{p}_{c_i}^* \forall i \in [|C|]$ which makes $\tilde{L}$ a tight upper bound of $f$.*

*Proof.* We begin by applying the smoothness assumption on the loss function $\ell_k^i$ for each class $i$. By Assumption 2, $\ell_k^i$ is $\beta_i$-smooth, which implies that for prompts $\mathbf{m}(k) \in \mathcal{P}$ and $\mathbf{p}_{c_i} \in \mathcal{P}$, for $k \in [n]$ and

$i \in [|C|]$ we have

$$\ell_k^i(\mathbf{m}(k)) \leq \ell_k^i(\mathbf{p}_{c_i}) + \nabla \ell_k^i(\mathbf{p}_{c_i})^\top (\mathbf{m}(k) - \mathbf{p}_{c_i}) + \frac{\beta_i}{2} \|\mathbf{m}(k) - \mathbf{p}_{c_i}\|^2, \tag{26}$$

$$\text{We define} \quad \beta_{\max} = \max_{i \in [|C|]} \beta_i, \tag{27}$$

which gives us

$$\ell_k^i(\mathbf{m}(k)) \leq \ell_k^i(\mathbf{p}_{c_i}) + \nabla \ell_k^i(\mathbf{p}_{c_i})^\top (\mathbf{m}(k) - \mathbf{p}_{c_i})) + \frac{\beta_{\max}}{2} \|\mathbf{m}(k) - \mathbf{p}_{c_i}\|^2. \tag{28}$$

Now we know that $\mathcal{P}$ is compact, let the diameter be $D := \sup_{x,y \in \mathcal{P}} \|\mathbf{x} - \mathbf{y}\|$ which gives us

$$\|\mathbf{p}_1 - \mathbf{p}_2\| \leq D \quad \text{for all } \mathbf{p}_1, \mathbf{p}_2 \in \mathcal{P} \tag{29}$$

$$\Rightarrow \|\mathbf{m}(k) - \mathbf{p}_{c_i}\| \leq D. \tag{30}$$

Since by Assumption 1 $l^i(\mathbf{p})$ is $\beta_i$-smooth, we have for $\mathbf{x}, \mathbf{y} \in \mathcal{P}$

$$\|\nabla \ell_k^i(\mathbf{x}) - \nabla \ell_k^i(\mathbf{y})\| \leq \beta_i \|\mathbf{x} - \mathbf{y}\| \tag{31}$$

$$\leq \beta_{\max} \|\mathbf{x} - \mathbf{y}\| \quad \text{From equation 27} \tag{32}$$

$$\text{If} \quad \forall \delta \geq 0 \quad \|\mathbf{x} - \mathbf{y}\| \leq \delta, \quad \text{for} \quad \epsilon = \delta \beta_{\max} \quad \text{we have}$$

$$\|\nabla \ell_k^i(\mathbf{x}) - \nabla \ell_k^i(\mathbf{y})\| \leq \epsilon \Rightarrow \|\nabla \ell_k^i(\mathbf{x})_j - \nabla \ell_k^i(\mathbf{y})_j\| \leq \epsilon, j \in [d] \tag{33}$$

$\nabla \ell_k^i(\mathbf{x})_j$ is a continuous mapping of compact metric space $\mathcal{P}$ into metric space $\mathbb{R}$

$$\Rightarrow \nabla \ell_k^i(\mathbf{x})_j \quad \text{is compact} \quad \forall j \in [d].$$

Let $B_{i_j}$ be the diameter of $\nabla \ell_k^i(\mathcal{P})_j$, $j \in [d]$, then

$$\|\nabla \ell_k^i(\mathbf{p}_{c_i})\| \leq B_i = \sum_{j=1}^{d} |B_{i_j}|. \tag{34}$$

Using Cauchy-Schwartz inequality and from 34 & 30 we have

$$\nabla \ell_k^i(\mathbf{p}_{c_i})^\top (\mathbf{m}(k) - \mathbf{p}_{c_i}) \leq D B_i \tag{35}$$

From 28

$$\ell_k^i(\mathbf{m}(k)) \leq \ell_k^i(\mathbf{p}_{c_i}) + \tilde{C}_k + \frac{\beta_{\max}}{2} \|\mathbf{m}(k) - \mathbf{p}_{c_i}\|^2, \tag{36}$$

where $\tilde{C}_k = D B_i$ . The global loss of the clients is given by

$$L = \frac{1}{n} \sum_{k=1}^{n} \left( \sum_{i=1}^{|C|} \delta_k^i \cdot \ell_k^i(\mathbf{m}(k)) \right) \tag{37}$$

$$\leq \tilde{L} = \frac{1}{n} \sum_{k=1}^{n} \left[ \sum_{i=1}^{|C|} \delta_k^i \left( \ell_k^i(\mathbf{p}_{c_i}) + \frac{\beta_{\max}}{2} \|\mathbf{m}(k) - \mathbf{p}_{c_i}\|^2 \right) \right] + \tilde{C}, \tag{38}$$

$$\text{where} \quad \tilde{C} = \frac{1}{n} \sum_{k=1}^{n} \tilde{C}_k, \tag{39}$$

which proves the first part of our main proposition 1 in the paper.

If $\mathbf{p}_{c_i} = \mathbf{p}_{c_i}^*$, we have a tight upper bound $\tilde{L} = \frac{1}{n} \sum_{k=1}^{n} \left[ \sum_{i=1}^{|C|} \delta_k^i \left( \ell_k^i(\mathbf{p}_{c_i}) + \frac{\beta_{\max}}{2} \|\mathbf{m}(k) - \mathbf{p}_{c_i}\|^2 \right) \right]$, because $\nabla \ell_k^i(\mathbf{p}_{c_i})$ vanishes, according to Assumption 3.

We are interested in finding the optimal client prompts $\mathbf{m}(k)$ for each client $k$.

$$\frac{\partial \tilde{L}_k}{\partial \mathbf{m}(k)} = \frac{1}{n} \sum_{i=1}^{|C|} \delta_k^i \beta_{\max} \left( \mathbf{m}(k) - \mathbf{p}_{c_i} \right) \tag{40}$$

$$= \frac{\beta_{\max}}{N} \left( \mathbf{m}(k) - \sum_{i=1}^{|C|} \delta_k^i \mathbf{p}_{c_i} \right), \quad \text{since} \sum_{i=1}^{|C|} \delta_k^i = 1 \tag{41}$$

$$\text{Setting} \quad \frac{\partial \tilde{L}_k}{\partial \mathbf{m}(k)} = 0, \text{ we have} \quad \mathbf{m}(k) = \sum_{i=1}^{|C|} \delta_k^i \mathbf{p}_{c_i} \tag{42}$$

which gives the second part of our proposition 1, and completes our proof. $\qquad\square$

### A.6.2 CCMP as MMSE estimator of the true class prompt

Here we clarify the details of the distribution $p_k(\mathbf{cls}_{l-1}, \mathbf{p}) = p_k(\mathbf{p}|\mathbf{cls}_{l-1}) p_k(\mathbf{cls}_{l-1})$ especially $p_k(\mathbf{cls}_{l-1})$. We define it as the density induced by the deterministic transformation of the data distribution through the preceding network layers.

Let $T_{l-1} : \mathcal{X} \to \mathbb{R}^d$ represent the composite non-linear mapping performed by the first $l-1$ layers of the ViT, such that for any input $\mathbf{x}$, the representation is given by $\mathbf{cls}_{l-1} = T_{l-1}(\mathbf{x})$. We can equip a probabiity space on the input as $(\mathcal{X}, \mathcal{F}, P)$, where $\mathcal{F}$ is a $\sigma$-algebra (typically the Borel $\sigma$-algebra) and $P$ is the probability measure on $\mathcal{X}$. We also equip the Measurable space of the $\mathbb{R}^d$ as $(R^d, \mathbb{B}(\mathbb{R}^d))$.

If $B$ is a Borel-measurable subset of $\mathbb{R}^d$. $\text{Pr}(B)$ is given by the pushforward measure of P on $B$ by $T_{l-1}$ which is $P(T_{l-1}^{-1}(B))$. We can always do this as the map $T_{l-1}$ is continous and hence measurable.

Consequently, $p_k(\mathbf{cls}_{l-1})$ is the probability density induced by this distribution derived via $P(\mathbf{x})$ under the mapping $T_{l-1}$. The posterior probability $p_k(\mathbf{p} = \mathbf{p}_{c_i}|\mathbf{cls}_{l-1})$ only implies that once we observe $\mathbf{cls}_{l-1}$ the probability that it belongs to a class $i$.

This is how we model the joint distribution $p_k(\mathbf{cls}_{l-1}, \mathbf{p})$.

We assume a joint data distribution $P(\mathbf{x}, y)$ over the input space $\mathcal{X}$ and the set of class labels $\mathcal{Y}$ with marginal $P(\mathbf{x})$. To formalize the notions, we define the input space as a probability space

**Proposition 2.** *If the cls tokens and the class-specific prompts at input of layer $l$ has the joint density given by $p_k(\mathbf{cls}_{l-1}, \mathbf{p})$ as in Eq. 17, then the CCMP prompt for a client $k$, $\mathbf{m}_{l-1}(k)$ obtained in Eq. 6 is Minimum Mean Squared Estimator (MMSE) of the true class prompt.*

*Proof.* Consider the following mean-squared error

$$J(\hat{\mathbf{p}}) = \mathbb{E}\|\mathbf{p} - \hat{\mathbf{p}}\|^2 \tag{43}$$

where the expectation is taken across the joint distribution of $p_k(\mathbf{p}, \mathbf{cls}_{l-1})$. The $\hat{\mathbf{p}}$ that's minimizes the $J(\hat{\mathbf{p}})$ is the MMSE estimator, and $\mathbf{p}$ is our true class prompt. We have the following

$$
\begin{aligned}
J(\hat{\mathbf{p}}) &= \mathbb{E}\|\mathbf{p} - \hat{\mathbf{p}}\|^2 \\
&= \mathbb{E}\|\mathbf{p} - \mathbb{E}[\mathbf{p}|\mathbf{cls}_{l-1}] + \mathbb{E}[\mathbf{p}|\mathbf{cls}_{l-1}] - \hat{\mathbf{p}}\|^2 \\
&= \mathbb{E}\|\mathbf{p} - \mathbb{E}[\mathbf{p}|\mathbf{cls}_{l-1}]\|^2 + \mathbb{E}\|\mathbb{E}[\mathbf{p}|\mathbf{cls}_{l-1}] - \hat{\mathbf{p}}\|^2 \\
&\quad + 2\mathbb{E}\langle \mathbf{p} - \mathbb{E}[\mathbf{p}|\mathbf{cls}_{l-1}], \mathbb{E}[\mathbf{p}|\mathbf{cls}_{l-1}] - \hat{\mathbf{p}} \rangle \\
&= \mathbb{E}\|\mathbf{p} - \mathbb{E}[\mathbf{p}|\mathbf{cls}_{l-1}]\|^2 + \mathbb{E}\|\mathbb{E}[\mathbf{p}|\mathbf{cls}_{l-1}] - \hat{\mathbf{p}}\|^2.
\end{aligned}
$$

The equality is obtained as the cross term is zero i.e we have $\mathbb{E}[\langle \mathbf{p} - \mathbb{E}[\mathbf{p}|\mathbf{cls}_{l-1}], \mathbb{E}[\mathbf{p}|\mathbf{cls}_{l-1}] - \hat{\mathbf{p}} \rangle] = 0$. It follows by using the iterated expectation as shown below.

$$
\begin{aligned}
\mathbb{E}\langle \mathbf{p} - \mathbb{E}[\mathbf{p}|\mathbf{cls}_{l-1}], \mathbb{E}[\mathbf{p}|\mathbf{cls}_{l-1}] - \hat{\mathbf{p}} \rangle &= \mathbb{E}[\mathbb{E}[\langle \mathbf{p} - \mathbb{E}[\mathbf{p}|\mathbf{cls}_{l-1}], \mathbb{E}[\mathbf{p}|\mathbf{cls}_{l-1}] - \hat{\mathbf{p}} \rangle | \mathbf{cls}_{l-1}]] & (44) \\
&= \mathbb{E}[\mathbb{E}[\langle \mathbf{p} - \mathbb{E}[\mathbf{p}|\mathbf{cls}_{l-1}] | \mathbf{cls}_{l-1}, \mathbb{E}[\mathbf{p}|\mathbf{cls}_{l-1}] - \hat{\mathbf{p}} \rangle]] & (45) \\
&= 0. & (46)
\end{aligned}
$$

We now have

$$
J(\hat{\mathbf{p}}) = \mathbb{E}\|\mathbf{p} - \mathbb{E}[\mathbf{p}|\mathbf{cls}_{l-1}]\|^2 + \mathbb{E}\|\mathbb{E}[\mathbf{p}|\mathbf{cls}_{l-1}] - \hat{\mathbf{p}}\|^2. \tag{47}
$$

From the above Eq. 47 it can be readily seen that $J(\hat{\mathbf{p}})$ is minimized by setting the value of $\hat{\mathbf{p}} = \mathbb{E}[\mathbf{p}|\mathbf{cls}_{l-1}]$

$$
\mathbb{E}[\mathbf{p}|\mathbf{cls}_{l-1}] = \sum_{m=1}^{|C|} p(\mathbf{p} = \mathbf{p}_{c_m}|\mathbf{cls}_{l-1})\mathbf{p}_{c_m}. \tag{48}
$$

From Eq. 16, we can rewrite the above equation

$$
\begin{aligned}
\mathbb{E}[\mathbf{p}|\mathbf{cls}_{l-1}] &= \sum_{m=1}^{|C|} s_{i,l-1,k}^m \cdot \mathbf{p}_{c_m} \\
&= \mathbf{P}_C * \mathbf{s}_{i,l-1,k}.
\end{aligned}
$$

From Eq. 25 we conclude that $\mathbb{E}[\mathbf{p}|\mathbf{cls}_{l-1}] = \mathbf{m}_{l-1}$. □

### A.6.3 Convergence

We assume the following assumptions on the loss functions based on (Karimireddy et al., 2020; Acar et al.).

**A 4.** *The loss functions $f_k$ are Lipschiltz smooth, i.e., $\|\nabla f_k(\boldsymbol{\theta}_1) - \nabla f_k(\boldsymbol{\theta}_2)\| \leq \beta\|\boldsymbol{\theta}_1 - \boldsymbol{\theta}_2\|$.*

**A 5.** $\frac{1}{n}\sum_{k\in[n]}\|\nabla f_k(\boldsymbol{\theta})\|^2 \leq G^2 + B^2\|\nabla f(\boldsymbol{\theta})\|^2$, *where $f(\boldsymbol{\theta}) = \frac{1}{n}\sum_{k\in[n]} f_k(\boldsymbol{\theta})$. This is referred to bounded gradient dissimilarity assumption,*

**A 6.** *let $\mathbb{E}\|\nabla l(\boldsymbol{\theta}, (x,y)) - \nabla f_k(\boldsymbol{\theta})\| \leq \sigma^2$, for all $k$ and $\boldsymbol{\theta}$. Here $l(\boldsymbol{\theta}, (x,y))$ is loss evaluated on the sample $(x,y)$ and $f_k(\boldsymbol{\theta})$ is expectation across the samples drawn from $\mathcal{D}_k$. This is a bounded variance assumption.*

In the above assumptions, the parameter $\boldsymbol{\theta}$ denotes the trainable, shared, and class-specific prompts along with the classification head parameters.

The entire computation of the soft scores $\mathbf{s}_{i,l-1,k}$ for the client $k$, based on `cls`, can be viewed as a part of the model architecture itself (Fig.1) and encapsulated inside the client's loss function.

We then have the following proposition.

**Proposition 3.** *Theorem V of Karimireddy et al. (2020) in Appendix D.2: let $\boldsymbol{\theta}^* = \arg\min_{\boldsymbol{\theta}} f(\boldsymbol{\theta})$, the global step-size be $\alpha_g$ and the local step-size be $\alpha_l$. When the update period $R$ is very large or $\tau >> 1$, the PEP-FedPT algorithm will have contracting gradients. If Initial model is $\boldsymbol{\theta}^0$, $F = f(\boldsymbol{\theta}^0) - f(\boldsymbol{\theta}^*)$ and for constant $M$, then in $T$ rounds, the model $\boldsymbol{\theta}^T$ satisfies $\mathbb{E}[\|\nabla f(\boldsymbol{\theta}^T)\|^2] \leq O(\frac{\beta M \sqrt{F}}{\sqrt{TLS}} + \frac{\beta^{1/3}(FG)^{2/3}}{(T+1)^{2/3}} + \frac{\beta B^2 F}{T})$.*

The above proposition states that the PEP-FedPT algorithm requires $\mathcal{O}(\frac{1}{\epsilon^2})$ communication rounds to make the average gradients of the global model smaller, i.e., $\mathbb{E}[\|\nabla f(\boldsymbol{\theta}^T)\|^2] \leq \epsilon$. The result is plug and play because we only employ global prompts and parameters for the training.

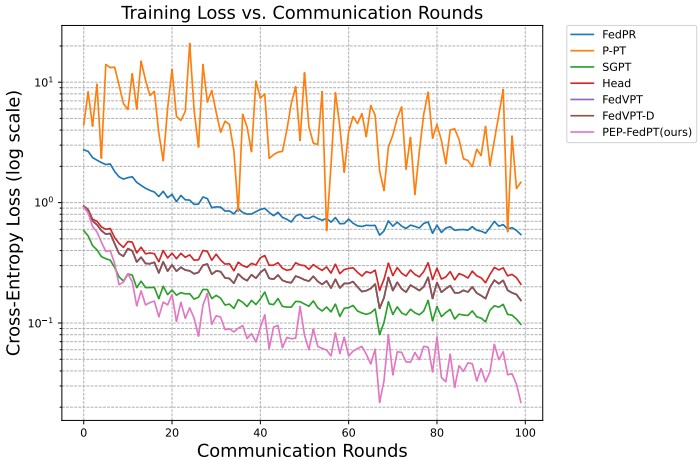

Figure 13: Comparison of training loss of various algorithms on CIFAR-100 dataset

Figure 13 illustrates the training loss (cross-entropy, log scale) versus communication rounds for various baselines. We observe that P-PT struggles to converge, while FedPR and Head show limited improvements with early plateauing. Methods such as SGPT and FedVPT achieve more stable convergence, and FedVPT-D further reduces the loss by incorporating additional regularization. In contrast, our proposed PEPFedPT consistently outperforms all baselines, achieving both faster convergence and the lowest final loss. Our theory, which minimizes the quadratic upper bound at convergence is expoected to have lower training loss. This empirical trend aligns with our predictions, thereby ensuring improved stability and convergence in practice.

### A.6.4 Analysis of CCMP when the scores are the function of data

We show that CCMP minimizes the quadratic upper bound on the loss even when the scores are functions of both the data and class-priors. We show an upper bound on the loss for a given input $x$. This helps us remove assumptions related to the temperature parameter and thus gives a more general theoretical analysis. Our notations are restated accordingly. We denote the estimate of class prompt for class $i$ at any round to be $\mathbf{p}_{c_i}$. We denote the class prompts by $\mathbf{P}_C = [\mathbf{p}_{c_1}, \mathbf{p}_{c_2} \ldots, \mathbf{p}_{c_{|C|}}]$. Let $\mathbf{m}(k, \mathbf{x})$ denote the prompt used at client $k$ for data point $\mathbf{x}$. [5], and let the total number of clients be $n$, and $\delta_k^i$ denote the empirical probability that a data point at client $k$ belongs to class $i$. We assume that the joint density of the data in client $p_k(\mathbf{x}, y)$ is modeled as $p_k(\mathbf{x}, y) := p_k(\mathbf{x})p_k(y|\mathbf{x})$, the posterior $p_k(y|\mathbf{x})$ is assumed to be given by the scores in Eq. 15 which we denote by $s_{k,i,\mathbf{x}}$ and we model $p_k(y = i|\mathbf{x})$ by defining $p_k(y = i|\mathbf{x}) := s_{k,i,\mathbf{x}}$. Let $\mathcal{P}$ be the set of all possible prompts across all the clients, such that $\mathbf{m}(k, \mathbf{x}) \in \mathcal{P}, \quad \forall k \in \{1, 2, \ldots, n\} \quad, \forall \mathbf{x}$. The overall loss of the client $k$ is denoted by the $\mathbb{E}[l_k(\mathbf{m}(k, \mathbf{x}), \mathbf{x}, y)]$. Note the expectation is over the $p_k(\mathbf{x}, y)$. The goal is to estimate $\mathbf{m}(k, \mathbf{x})$ as a function of class prompts $\{\mathbf{p}_{c_1}, \mathbf{p}_{c_2} \ldots, \mathbf{p}_{c_{|C|}}\}$. The global loss across all clients can be computed as $f = \frac{1}{n} \sum_{k=1}^{n} \mathbb{E}[l_k(\mathbf{m}(k, \mathbf{x}), \mathbf{x}, y)]$.

We now state the following assumptions:

---

[5]for notation convenience, we drop the layer index $j$ from $\mathbf{m}_j(k, x)$.

**A 7.** $\mathcal{P}$ *is compact subset of* $\mathbb{R}^d$, *where d is the token dimension.*

**A 8.** $l_k(\boldsymbol{\theta}, \mathbf{x}, y)$ *is* $\beta$ *smooth in argument* $\boldsymbol{\theta}$ *with parameter* $\beta$ $\forall y \in [|C|], \forall \mathbf{x}, \forall k \in [n]$.

**Proposition 4.** *If* $\ell_k(\boldsymbol{\theta}, (\mathbf{x}, y))$ *satisfies the above assumptions 7 to 8, we show that overall loss function* $f = \frac{1}{n} \sum_{k=1}^{n} \mathbb{E}[\ell_k(\mathbf{m}(k, \mathbf{x}), (\mathbf{x}, y))]$ *can be upper bounded as* $f \leq \tilde{L} = \frac{1}{n} \sum_{k=1}^{n} \mathbb{E}\left[\sum_{i=1}^{|C|} s_k^i \left(\ell_k^i(\mathbf{p}_{c_i}) + \frac{\beta_{\max}}{2} \|\mathbf{m}(k) - \mathbf{p}_{c_i}\|^2\right)\right] + \tilde{C}$ *and it is minimized at* $\mathbf{m}(k) = \sum_{c=1}^{|C|} s_k^i \mathbf{p}_{c_i}$, $\forall k \in [n]$. *which is equivalent to the (CCMP) described in sec.4.2 .* $\tilde{C}$ *is a constant which depends on* $\mathcal{P}$. *The* $\mathbb{E}$ *is over the distribution of the data* $\mathbf{x}$. *Here we defined* $\mathbf{m}(k) := \mathbf{m}(k, \mathbf{x})$, $\ell_k^i(\boldsymbol{\theta}) := \ell_k(\boldsymbol{\theta}, \mathbf{x}, y = i)$ *and* $s_k^i := s_{k,i,\mathbf{x}}$

*Proof.* we expand the clients loss $\mathbb{E}[\ell_k(\mathbf{m}(k, \mathbf{x}), \mathbf{x}, y)]$ as below

$$\mathbb{E}[\ell_k(\mathbf{m}(k, \mathbf{x}), \mathbf{x}, y)] = \mathbb{E}[\mathbb{E}[\ell_k(\mathbf{m}(k, \mathbf{x}), \mathbf{x}, y)]|\mathbf{x}] \tag{49}$$

$$= \mathbb{E}[\sum_{i=1}^{|C|} \ell_k(\mathbf{m}(k, \mathbf{x}), \mathbf{x}, y = i) p_k(y = i|\mathbf{x})] \tag{50}$$

$$= \mathbb{E}[\sum_{i=1}^{|C|} \ell_k(\mathbf{m}(k, \mathbf{x}), \mathbf{x}, y = i) s_{k,i,\mathbf{x}}] \tag{51}$$

$$= \mathbb{E}[\sum_{i=1}^{|C|} \ell_k^i(\mathbf{m}(k)) s_k^i]. \tag{52}$$

In the last step we use the definitions in the proposition i.e, $\mathbf{m}(k) := \mathbf{m}(k, \mathbf{x})$, $\ell_k^i(\mathbf{m}(k)) := \ell_k(\mathbf{m}(k, \mathbf{x}), \mathbf{x}, y = i)$ and $s_k^i := s_{k,i,\mathbf{x}}$.

If the Lipschitz smooth(8) and compactness(7) assumptions hold, then by following similar arguments from 26 till 38 we will have the global loss of clients given by,

$$f = \frac{1}{n} \sum_{k=1}^{n} \mathbb{E}\left[\sum_{i=1}^{|C|} s_k^i \cdot \ell_k^i(\mathbf{m}(k))\right] \tag{53}$$

$$\leq \tilde{L} = \frac{1}{n} \sum_{k=1}^{n} \mathbb{E}\left[\sum_{i=1}^{|C|} s_k^i \left(\ell_k^i(\mathbf{p}_{c_i}) + \frac{\beta}{2} \|\mathbf{m}(k) - \mathbf{p}_{c_i}\|^2\right)\right] + \tilde{C}, \tag{54}$$

which proves the first part.

We are interested in finding the optimal client prompts $\mathbf{m}(k)$ for each client $k$ and for each data point $\mathbf{x}$. This is obtained by optimizing the argument inside the expectation which is $\left[\sum_{i=1}^{|C|} s_k^i \left(\ell_k^i(\mathbf{p}_{c_i}) + \frac{\beta}{2} \|\mathbf{m}(k) - \mathbf{p}_{c_i}\|^2\right)\right]$ with respect to $\mathbf{m}(k)$.

$$\frac{\partial \sum_{i=1}^{|C|} s_k^i \left(\ell_k^i(\mathbf{p}_{c_i}) + \frac{\beta}{2} \|\mathbf{m}(k) - \mathbf{p}_{c_i}\|^2\right)}{\partial \mathbf{m}(k)} = \frac{1}{n} \sum_{i=1}^{|C|} s_k^i \beta (\mathbf{m}(k) - \mathbf{p}_{c_i}) \tag{55}$$

$$= \frac{\beta}{N} \left(\mathbf{m}(k) - \sum_{i=1}^{|C|} s_k^i \mathbf{p}_{c_i}\right), \quad \text{since} \sum_{i=1}^{|C|} s_k^i = 1 \tag{56}$$

$$\text{Setting } \frac{\partial \tilde{L}_k}{\partial \mathbf{m}(k)} = 0, \text{ we have} \quad \mathbf{m}(k) = \sum_{i=1}^{|C|} s_k^i \mathbf{p}_{c_i} \tag{57}$$

which gives the second part of our proposition 1, and completes our proof. $\square$

