# OpenReview forum: "Prompt Estimation from Prototypes for Federated Prompt Tuning of Vision Transformers"
_TMLR — Accepted by TMLR_

### Review · Reviewer_AyXx · 2025-11-21

**Summary Of Contributions:**

The authors propose an efficient training strategy for the ViT architecture in an FL setting with heterogeneous clients. Here, the authors adapt the visual prompt tuning method to the FL setting by adding class-specific prompts as training parameters. The authors mix class-specific prompts for each client based on their closeness to a global class prototype, yielding class contextualized mixed prompts. The authors show that their mixing strategy achieves the optimal error under the assumption on the type of client heterogeneity (Proposition 1). Additionally, under the assumption of a "well-specified" model (Eq. (16)), the authors show that CCMP prompts are optimal estimators of class prompts (Proposition 2). The authors show an advantage of their method in several class- and domain-heterogeneous settings.

**Strengths**
1. Motivation is clear.
2. The authors provide a theoretical justification for some elements of the method.
3. Strong performance in the proposed evaluation.

**Weaknesses**
1. I find Section 4.1 quite confusing.
2. I think the authors do not sufficiently discuss the limitations of their approach.

**Audience:**

Yes

**Audience Explanation:**

I think this paper could be interesting to people working on applying ML in the medical domain, since here FL seems like a promising technique for private model optimization.

**Broader Impact Concerns:**

I don't see a need for a Broader Impact Statement.

**Claims And Evidence:**

Yes

**Claims Explanation:**

I think the evidence is sufficient. However, I believe the limitations and implicit assumptions of the proposed approach should be further discussed.

1. The method implicitly assumes that it deals only with clients that differ only in the distribution of classes across them. It can be seen from the construction in Section 4.1, where the only way of model personalization is the mixing of class prompts. I believe this assumption should be made explicit and discussed as one of the method's limitations.
2. The assumptions in Section 4.3 lack a semantical explanation. In particular, Assumption A 2 in Section 4.3.1 again seems to imply a specific form of clients' heterogeneity. Similarly, Eq. (16) looks quite specific and implicitly assumes a lot about the outcome of optimization. I believe these assumptions should be discussed more thoroughly.

**Requested Changes:**

1. Please make Section 4.1 clearer. In particular, explain for which layers we need to add CCMP and what happens before and after this addition.
2. Please clarify the scope and limitations of your work regarding client heterogeneity.
3. Please discuss the assumptions in Section 4.3.

---

> ### Author Response · Authors · 2026-01-28
> **Response to Reviewer AyXx**
>
> We thank the reviewer for evaluating our work. We now address the concerns.
>
> **Regarding Section 4.1**
>
> The motivation behind our approach can be explained as follows:
>
>  1) The class priors help capture the degree of label heterogeneity.
>
>  2) The cls-tokens are responsible for capturing the feature heterogeneity in images  within the same class.
>
> The composite score together therefore, gives a refined estimate of weights for prompt mixing. Even if the class distribution was uniform across all the clients, the method would still give a reasonable estimate because  of the dependence of CCMP scores on proximity between centroids and cls-tokens.  Suppose, if the cls-tokens did not capture any feature heterogeneity, the overall prompt mixing weights would be uniform, leading to the collapse of all the class prompts to a single prompt, and we expect the performance to be similar to VPT. We do not see this happen, for example, consider the DomainNet experiment, FedVPT attains a mean Acc of 84.23 (Table 2 of the main paper). Our method without class priors attains a Mean Acc of 86.34 (Table A.4.2). This performance improvement is the empirical evidence that similarity scores between clients' cls-token and class centroids indeed provide personalization.
>
> **On the layers where CCMP is added**
>
> CCMP are added at layers 5,6,7. In Table 12, we performed the analysis of our method, PEP-FedPT. It can be seen that adding the CCMP prompts too early in the ViT is not beneficial, as the cls token representations at very early layers do not have better representations. Adding the prompts at later layers is also not beneficial, even though the cls tokens have better representations, since the prompts inserted are not deep enough to learn something useful.
>
>
> **On the Assumptions**
>
> Assumption (A1) states that the prompt space $\mathcal{P}$ is compact, which means that all the possible prompt values lie within a bounded region and they do not drift off to infinity.
>
> Assumption (A2) states that the loss function for each class i is Lipschitz smooth with parameter $\beta_i$. We would like to point out that it does not assert any additional assumption on heterogeneity. The loss function $l_k$ for each client has been broken down and written in terms of each class i for the sake of convenience. We merely write $f_k(\theta_k,y=i)$  as $l^i_k$
>
> Assumption (A3) highlights that each class i has a corresponding true class prompt $\mathbf{p}^*_{c_i},$ for which $f_k(\theta_k,y=i)$ achieves the minimum value.
>
> Eq 16. only implies the modeling of our estimation of probability with which to insert a certain class prompt, when the cls-token at the preceding layer has already been observed.
>
> **Discussion on the  Limitations**
>
> The method cannot be trivially extended to Semi-supervised settings, as mentioned in the paper. In the case of long-tail distributions, the method depends on the ability to compute class centroids and the availability of priors. In the long tail distributions, the scarcity of certain classes can impact the quality of centroids and might not yield the best performance.
>
> The method exploits the non-iid nature of the data across clients to achieve personalization via global parameters. If the data is iid, the method's utility is limited, as it collapses to the usual Fed-VPT.
>
> *As the reviewer suggested, we have accommodated all the suggested changes into the main draft. We will update the draft very soon once we accommodate all the suggested changes from all reviewers.*

---

> > ### Comment · Reviewer_AyXx · 2026-01-29
> >
> > I appreciate the authors' clarifications. However, some of my concerns remain unaddressed.
> >
> > **Regarding Section 4.1**
> >
> > The motivation behind the approach is now clearer to me. However, I still think the authors should comment on the method's potential limitations (even though the method achieves good empirical performance).
> >
> > **On the layers where CCMP is added**
> >
> > Here, my comment was about the clarity of the presentation. Eq. (4) suggests that CCMP is added at every layer, while Eq. (7) suggests that it is added to some layer j (and maybe layers after). Without clarification about adding prompts only to the middle layers, it is hard to understand the overall optimization problem that the authors are trying to solve.
> >
> > **On the Assumptions**
> >
> > I completely understand the mathematical meaning of the assumptions. However, the semantic meaning is still elusive to me. From what I understand, A3 directly constrains the considered heterogeneity regimes since the optimal class prompts are the same for each client, suggesting the lack of feature heterogeneity.
> >
> > Regarding Eq. (16) and Section 4.3.2 in general, I do not fully understand the scope of the analysis. First, I do not understand what probability events give rise to $p_k(\mathbf{cls}_{l-1})$. Second, if we indeed define posterior probability using Eq. (16), I do not fully understand what kind of real-world situation this assumption describes. Currently, this section reads like a mathematical exercise with no connection to the method's real-world efficiency (i.e., the authors prove the method's optimality under the assumption that it is optimal).
> >
> > **Discussion on the Limitations**
> >
> > I think the discussion of limitations should also address the theoretical limitations of the analysis and the types of heterogeneity distributions the method is designed to tackle (e.g., as I see it, the method is not well-suited to concept drift heterogeneity).

---

> ### Author Response · Authors · 2026-01-30
> **response to reviewer AyXx**
>
> We thank the reviewer for the feedback and giving the opportunity to clarify the concerns
>
> **On the layers where CCMP is added**
>
> Eq. 4 is the generic equation to show the input and output when CCMP is inserted. Eq 7 restates the same while elaborating. However, we understand that it may be confusing, so in the modified draft of our paper, we have changed Eq.4 to represent just the  generic input-output of a ViT layer upon prompt insertion, and it has been more methodically organized.
>
> **On Assumptions**
>
> We agree that (Assumption 3) assuming the gradient to be zero for a single class prompt across all clients rules out feature heterogeneity, while it does still account for label heterogeneity. We will explicitly mention this as one of the limitations in the paper. However, we would like to bring to attention that the general quadratic upper bound that we have on the global loss function does not depend on this particular assumption. It is only required to make the bound tight at the end of the proof. We have updated the meaning of the Assumption 3 and clearly mentioned it in proposition 1 in the revised draft.
>
> **On the meaning of  $p_k({cls}_{l-1})$**
>
> To formally clarify the origin of $p_k({cls}_{l-1})$, In short, we define it as the density induced by the deterministic transformation of the data distribution through the preceding network layers. The details are below.
>
> Let $T_{l-1}: \mathcal{X} \to \mathbb{R}^d$ represent the composite non-linear mapping performed by the first $l-1$ layers of the ViT, such that for any input $\mathbf{x}$, the representation is given by $\mathbf{cls}_{l-1}$
>
>  which is defined to be $T_{l-1}(\mathbf{x})$
>
> We can equip a probabiity space on the input as  $(\mathcal{X}, \mathcal{F}, P)$, where $\mathcal{F}$ is a $\sigma$-algebra (typically the Borel $\sigma$-algebra) and $P$ is the probability measure on $\mathcal{X}$. We also equip the Measurable space of the $\mathbb{R}^d$  as   $({\mathbb{R}^d}, \mathbb{B}(\mathbb{R}^d))$.
>
> If  $B$ is a Borel-measurable subset of  $\mathbb{R}^d$. Pr($B$)  is given by the pushforward measure of
> P on $B$ by
> $T_{l-1}$ which is $P(T^{-1}_{l-1}(B))$.
>
> We can always do this as the map $T_{l-1}$ is continuous and hence measurable.
>
> Consequently,
>
> $p_k({cls}_{l-1})$
>
>  is the probability density induced by this distribution derived via $P(\mathbf{x})$ under the mapping $T_{l-1}$.
>
> The posterior probability $p_k(p|cls_{l-1})$ only implies that once we observe
> ${cls}_{l-1}$  the probability that it belongs to a class $i$.
>
> This is what we have modelled by our prediction according to Eq.15. We do not make any optimality assumption here; we simply model the joint density of
>
> $p_k({cls}_{l-1},p)=$
>
>  $p_k(cls_{l-1})$  $p_k(p|cls_{l-1}) $
>
> It is precisely under this modelling assumption that we interpret CCMP as MMSE optimal and did not claim anything more.
>
> We have added this discussion in the Sec.A.6.2 of the revised draft.
>
> **Limitations**
>
> The limitations due to Assumption 3 have been clearly mentioned in the Limitations section.

---

> ### Author Response · Authors · 2026-01-30
> **response to reviewer AyXx**
>
> **changes summary**
>
> *We have colored all the changes requested by reviewer AyXx in cyan*
>
>  Refined the method Section 4.1 (pages 5, 6)
>
> Clarification on the assumptions, especially A.3 (page 8)
>
> Refined the statement of the proposition to accommodate the impact of assumption (A.3) (page 8)
>
> We clearly state the limitation and impact of our assumption A.3 on the analysis in the limitation section (page 12)
>
> The detailed explanation of the modelling of $p_k(cls_{l-1},p)$ is provided in the Aec.A.6.2 of the appendix (page 29)

---

### Review · Reviewer_rvUd · 2026-01-06

**Summary Of Contributions:**

This paper proposes PEP-FedPT (Prompt Estimation from Prototypes for Federated Prompt Tuning), a novel federated learning framework designed for fine-tuning Vision Transformers (ViTs) in heterogeneous data settings. The key contributions are:
1. Unified Prompt Framework: Introduces a combination of shared prompts (global) and class-contextualized mixed prompts (CCMP) to balance generalization and personalization without storing client-specific parameters.
2. CCMP Mechanism: Dynamically mixes class-specific prompts using soft weights derived from global class prototypes and local class priors, enabling per-sample personalization using only globally shared information.
3. Theoretical Grounding: Shows that CCMP minimizes a quadratic upper bound on loss and is optimal in the minimum mean squared error (MMSE) sense.
4. Strong Empirical Results: Demonstrates superior performance across multiple datasets (CIFAR-100, TinyImageNet, DomainNet, iNaturalist) under label and feature heterogeneity, outperforming state-of-the-art baselines in both participating and held-out client evaluations.
5. Efficiency: Achieves high accuracy with significantly reduced communication and computation costs compared to existing methods.

Key Strengths:
1. Effectively addresses the personalization-generalization trade-off in federated prompt tuning.
2. Lightweight and scalable, with minimal added computational overhead.
3. Well-supported by both theoretical analysis and extensive experiments.

Key Weaknesses:
1. Relies on labeled data for estimating class priors and prototypes, limiting applicability to semi-supervised or unsupervised FL settings.
2. Privacy guarantees are not formally proven; relies on empirical measures like differential privacy.

**Audience:**

Yes

**Audience Explanation:**

The paper sits at the intersection of several active research areas: federated learning, vision transformers, parameter-efficient tuning, and personalized learning. The proposed method addresses a critical challenge in FL—balancing personalization and generalization—while being computationally efficient. Researchers and practitioners in FL, computer vision, and efficient deep learning would find the approach novel and practically relevant.

**Broader Impact Concerns:**

No major ethical concerns are raised by this work. The method is designed for efficient and privacy-aware federated learning, which aligns with ethical ML practices. However, as with any FL method, there is a risk of unintended bias propagation if data heterogeneity reflects societal biases. The authors should consider adding a Broader Impact Statement discussing:
1. Potential biases in class prototypes if client data distributions are skewed.
2. The importance of ensuring fairness across clients in personalized FL systems.
3. The environmental impact of reduced communication rounds as a positive contribution.

**Claims And Evidence:**

Yes

**Claims Explanation:**

Comprehensive experiments across four diverse datasets under varying heterogeneity settings (pathological, Dirichlet, feature imbalance).
Clear ablation studies that validate the contributions of shared prompts, CCMP, and class priors.
Theoretical proofs provided in the appendix, linking CCMP to loss minimization and MMSE optimality.
Comparative analysis against strong baselines, including personalized and global FL methods, with consistent outperformance.
Efficiency analysis showing reduced communication and computation costs.

**Requested Changes:**

1. Clarify Privacy Implications: While differential privacy (DP) is briefly discussed in Appendix A.4.1, the privacy-risk trade-off should be more explicitly addressed in the main paper, especially since prototypes are shared.
2. Expand on Limitations: The limitation regarding reliance on labeled data should be elaborated, suggesting potential extensions to semi-supervised or self-supervised FL as future work.
3. Visualization Enhancements: More figure could be better annotated to clearly show how class prompts and representations evolve across layers.
4. Reproducibility: Provide clearer hyperparameter settings and potentially release code upon publication.
5. The article mentions the usage of prompt in vision tasks, which can refer to the article " How to Design or Learn Prompt for Domain Adaptation?". This article proposes image encoder tuning, named Im-Tuning, for more separable image features.

---

> ### Author Response · Authors · 2026-01-28
> **Response to Reviewer rvUd**
>
> We thank the reviewer for the detailed feedback and address the raised questions below.
>
> **Privacy Clarifications**
>
> We acknowledge that our method requires the transmission of class prototypes to facilitate adaptive prompt mixing. However, these transmitted statistics represent aggregated, intermediate layers', low-dimensional summaries of the local data rather than raw data. As mentioned in [1], it is usually difficult for the server to extract sensitive data from category-level feature statistics alone (in our case, cls-prototypes). Moreover, in practical scenarios, even a gradient/model sharing algorithm like FedAvg would require a privacy-preserving technique for the system to be completely reliable. We will expand the discussion on the DP where we show that $(\epsilon,0)$ privacy can be attained in the revised draft.
>
> **Expand on limitation**
>
> The method cannot be trivially extended to Semi-supervised settings, as mentioned in the paper. In the case of long-tail distributions, the method depends on the ability to compute class centroids and the availability of priors. In the long tail distributions, the scarcity of certain classes can impact the quality of centroids and might not yield the best performance. The method exploits the non-iid nature of the data across clients to achieve personalization via global parameters. If the data is iid, the method's utility is limited, as it collapses to the usual Fed-VPT.
>
> **Visualization Enhancements**
>
> We have included a figure (Figure 9 in Appendix A.5.4)  across layers in the original draft. However, we have added the evolution of class prompts as training progresses in the latest draft of the paper.
>
> **Reproducibiity**
>
> Yes, we shall release the source code and include the link in the final version.
>
> **HyperParameter Detais**
>
> In Section A.3.2  of the original draft, we have clearly discussed the hyperparameter settings. We also add the following table that explain all the configurations
> | Dataset |  Classes | Clients | Classes / Client | Comm. Rounds | Participation Rate | Local Epochs | Centroid Update Interval |
> |----------------|-----------|-----------|----------------------------------|--------------|--------------------|--------------|--------------------------|
> | CIFAR-100 | 100 | 100 | 10 (Pathological / Dir(0.3)) | 100 | 5% | 5 | 10 |
> | Tiny-ImageNet | 200 | 200 | 10 (Pathological / Dir(0.3)) | 100 | 2.5% | 5 | 10 |
> | DomainNet | 10 | 60 | 5 | 50 | 10% | 5 | 10 |
> | iNaturalist | 1203 | 1018 | 10 | 500 | 1% | 2 | 10 |
>
>
> We will accommodate the mentioned reference in the final draft.
>
> *All the suggested changes will be accommodated in the final draft. We will update the draft at the earliest once we have incorporated all the reviewers' suggestions.*
>
> [1] Personalized federated learning with feature alignment and classifier collaboration. ( ICLR, 2023)

---

> ### Author Response · Authors · 2026-01-30
> **Response to Reviewer rvUd**
>
> **changes summary**
>
> *We have colored all the changes asked by reviewer rvUd in blue*
>
> We have added the discussion the privacy on page 7
>
> Related work based on prompt tuning for DA has been added on page 3
>
> Expanded the discussion on the limitations. (page 8)
>
> The complete hyper-parameter settings have been added in Sec A.3.2. (page 19)
>
> Expanded the discussion on the DP in Sec A.4.1. (page 20)
>
> Added Evolution of class-prompts representation across rounds in Sec.A.5.9. (page 27)

---

### Review · Reviewer_72NJ · 2026-01-20

**Summary Of Contributions:**

## Summary

This paper studies **federated visual prompt tuning (VPT)** for pre-trained **Vision Transformers (ViTs)** under **client heterogeneity**, focusing on the tension between (i) *global prompt tuning* (better generalization but weaker personalization) and (ii) *personalized prompt tuning* (strong local fit but poor generalization / scalability).

The proposed method, **PEP-FedPT (Prompt Estimation from Prototypes for Federated Prompt Tuning)**, introduces:

* **Shared prompts** ($P_S$) that are inserted at the input and are globally shared.
* **Class-specific prompts** ($P_C = [p_{c_1},\dots,p_{c_{|C|}}]$) that are also globally shared.
* A **Class-Contextualized Mixed Prompt (CCMP)** (m) constructed **per client / per sample** by mixing the class prompts using **soft weights** derived from:

  * similarity between a sample’s **cls token** representation at a chosen layer and **global class prototypes** aggregated on the server, and
  * the client’s **local class prior**.

The paper provides:

* A method to compute and aggregate **class prototypes** across clients over rounds (with a momentum-style update),
* A **theory section** arguing CCMP is optimal under certain assumptions (quadratic upper bound minimization and an MMSE-style argument),
* Extensive experiments on **CIFAR-100 / Tiny-ImageNet** (label heterogeneity) and **DomainNet / iNaturalist** (feature/domain heterogeneity), including a **held-out client evaluation**, ablations, and a computation/communication comparison.

## Key strengths

* **Clear and fairly intuitive problem setup** and notation in the preliminaries and method sections (overall, the formulation is easy to follow).
* **Strong empirical performance** across multiple datasets and heterogeneity settings (Tables 1–3; Appendix held-out table), with consistent gains in “mean accuracy” metrics.
* **Figure 2** is a genuinely interesting diagnostic: it highlights how predictive power varies across ViT layers, motivating where to insert CCMP.
* Helpful ablations on:

  * **Shared vs Shared+CCMP** (Table 4; Appendix Table 9),
  * **Impact of class priors** (Table 5; Appendix Table 8),
  * **Layer placement of CCMP** (Appendix Table 12).
* Includes a communication/computation discussion (Table 6) and additional appendix analyses.

## Key weaknesses / concerns

* **Evaluation metric consistency issues** (notably for iNaturalist “Worst Acc”) and **unclear fairness/interpretation** of held-out evaluations for baselines like pFedPG.
* **Theory–practice mismatch around the temperature ($\tau$)**: theory relies on ($\tau \gg 1$) but experiments fix ($\tau = 0.05$); there is no sensitivity analysis.
* **Heterogeneity robustness is under-explored**: the Dirichlet label-imbalance setting is shown for essentially **one** concentration value (0.3), despite the paper positioning itself as addressing heterogeneity broadly.
* **Clarity/notation problems**: the paper uses **($\delta$)** for both class priors and (in the appendix) the Dirichlet parameter, and there are places where the prompt-injection description/pseudocode is hard to implement directly.

**Additional Comments:**

Please refer to requested changes.

**Audience:**

Yes

**Audience Explanation:**

Federated adaptation of large foundation models (here, ViTs) using **parameter-efficient tuning** is an active and practically important topic. This submission addresses a real and well-known tension in FL (**generalization vs personalization under heterogeneity**) and proposes a mechanism (prototype-guided prompt mixing) that is simple enough to be broadly useful if clarified and validated. The combination of:

* strong empirical results,
* a prototype-based mixing strategy,
* and a focus on prompt tuning in FL

 should interest readers working on FL, PEFT/prompting, and robust generalization.

**Claims And Evidence:**

No

**Claims Explanation:**

The empirical evidence for “PEP-FedPT performs strongly on the reported benchmarks” is largely supported by the reported tables/plots. However, several claims and supporting evidence are **not yet consistent/clear enough** for me to answer “Yes” overall:

1. **Metric definition inconsistency (high impact on evidence clarity).**
   In the main paper, “Worst Local Accuracy” is defined as the worst-performing client (Evaluation Methodology). But in the appendix, the paper states that for **iNaturalist** the “Worst Acc” is instead **the minimum accuracy over the final 250 rounds**, and also notes that “worst client accuracy is zero for all methods” due to participation issues. This makes the meaning of the iNaturalist “Worst Acc” column unclear and potentially incomparable to other datasets/methods.

2. **Held-out evaluation vs. baseline applicability (fairness/interpretation).**
   Table 3 shows extremely low held-out “Testing Acc” for **pFedPG** (e.g., single-digit accuracy), which is consistent with the *expected behavior* of certain personalized methods under a “new clients, no adaptation” protocol—but then the comparison needs to be framed explicitly as such, and/or baselines should be given a fair adaptation strategy. Right now the evidence does not clearly separate “method is bad” from “method is not designed for this held-out protocol”.

3. **Theory–experiment mismatch for ($\tau$).**
   The theoretical claims tie the CCMP form to regimes like ($\tau \gg 1$), while experiments fix ($\tau = 0.05$). Without sensitivity analysis or an argument that the theoretical form is robust to very small ($\tau$), the theoretical evidence is not convincingly connected to the empirical configuration.

4. **Some quantitative statements are slightly inaccurate.**
   Example: the text reports an improvement over pFedPG on CIFAR-100 pathological of 2.57%, but Table 1’s mean accuracies imply 95.46 − 92.96 = 2.50%. This is minor but reinforces the need for careful verification.

Given these issues, I believe the core idea is promising and the empirical results look strong, but the **evidence is not yet fully accurate/convincing/clear**.

**Requested Changes:**

## Major changes

### 1) Clarify the held-out evaluation protocol and baseline fairness (especially pFedPG)

* Clearly specify: for “Testing Acc” on held-out clients, **are methods allowed any adaptation** (e.g., local fine-tuning of head/prompts, few-shot tuning, or any client-side steps), or is it **zero-adaptation**?
* If it is a **zero-adaptation** protocol, explicitly state that and discuss which baselines are applicable under this setting.
* The extremely low pFedPG held-out accuracies suggest either (a) the baseline is evaluated in a setting it is not designed for, or (b) there is an implementation/config mismatch. Either way, this requires explicit clarification and discussion.

###  2) Fix metric definitions and ensure consistent reporting (esp. iNaturalist “Worst Acc”)

* Align the definition of “Worst Acc” across datasets, or rename metrics so they are unambiguous.
* If iNaturalist “Worst Acc” is **min accuracy over the final 250 rounds** (as stated in the appendix), then:

  * label it accordingly in the main paper and Table 2 caption, and
  * ideally report an additional robustness metric that matches the “worst client” concept (e.g., worst over participating clients, or a lower quantile like 5th percentile).

### 3) Provide a sensitivity analysis on the temperature parameter ($\tau$)

* The paper uses ($\tau = 0.05$) for all experiments but theoretical statements relate CCMP properties to ($\tau \gg 1$).
* Please add an ablation varying ($\tau$) (e.g., log-scale sweep) and show:

  * performance stability,
  * behavior of the soft weights (entropy/sharpness),
  * and whether the “best” layer choices change.
* Alternatively (or additionally), revise the theoretical discussion to match the regime used in practice.

### 4) Vary the Dirichlet heterogeneity parameter (not only 0.3)

* The Dirichlet label-heterogeneity results use essentially one concentration value (0.3).
* Since the paper positions itself as addressing heterogeneity, please include results for multiple concentration values (e.g., ($\alpha \in {0.1,0.3,0.5,1.0}$) or similar) to demonstrate robustness of performance and the personalization/generalization tradeoff.

## Minor changes

### 1) Add a 2D personalization–generalization tradeoff plot

A 2D plot could provide a clearer picture of the tradeoff across methods. For example:

* x-axis: personalization,
* y-axis: generalization,
* each method as a point (or Pareto front).
  This would make the personalization/generalization behavior immediately interpretable and may further highlight the proposed method’s strengths.

### 2) Make the early overview more intuitive / higher-level

* The introduction currently has a text description, but it would benefit from a more “first-glance” explanation of the design choices and motivation:

  1. why cls-token prototypes,
  2. why combine with class priors,
  3. why insert at intermediate layers (and how layer choice was decided).
* Consider adding a simplified high-level summary or a short step-by-step algorithmic summary early (before deep notation), so readers can quickly form the mental model.

### 3) Expand related work with Federated Domain Generalization

Since the paper frames itself around generalization under distribution shift/heterogeneity, it would be beneficial to include a small subsection on **Federated Domain Generalization (FDG)** beyond foundation model prompt tuning, e.g.:

* “FedSR: A simple and effective domain generalization method for federated learning” (NeurIPS 2021)
* “Federated domain generalization with generalization adjustment” (CVPR 2023)
* “Federated unsupervised domain generalization using global and local alignment of gradients” (AAAI 2025)

  This would better position the contribution in the broader literature and make the paper more self-contained.

---

> ### Author Response · Authors · 2026-01-30
> **Response to Reviewer  72NJ (1)**
>
> We thank the reviewer for the detailed feedback and we now calrify the questions.
>
> **Metric definition inconsistency**
>
> Thank you for pointing out the ambiguity in the definition of “Worst Acc.” We agree that the metric should be consistently interpreted across datasets.
> For the iNaturalist experiments, we do not report worst-client accuracy. Under this experimental setup, the worst-client accuracy is zero for all methods, making the metric uninformative and potentially misleading. This behavior is caused by
>
>  (i) a very low client participation rate (1%), which results in some clients being sampled extremely infrequently, and
>
>  (ii) the presence of more than 150 clients with very small test sets (fewer than 10 samples).
>
>  For such clients, misclassification of only a few samples leads to a measured accuracy of zero, regardless of the underlying model quality. As a result, the worst-client metric is dominated by these degenerate cases rather than reflecting meaningful robustness or generalization behavior. To avoid ambiguity and ensure metric consistency, we remove the “Worst Acc” column for iNaturalist from the main table and explicitly state this rationale in the main text. Instead, as suggested by the reviewer, we report the 15% (percentile) accuracy, which provides a more stable and informative robustness measure by summarizing performance over the lower tail of the client accuracy distribution.
>
> *INaturalist metric clarification is provided on page (10)*
>
> We perform the following experiment with regard to the worst client accuracy; we provide the mean (std) over three runs.
>
> **Table: percentile accuracy on iNaturalist**
> | Method | 5% | 10% | 15% |
> |------|----|-----|-----|
> | Head | 0 | 12.50  (0.32) | 20.56  (0.47) |
> | VPT | 0 | 14.28  (0.16) | 23.21  (0.55) |
> | VPT-D | 10.50 (0.20) | 20.00 (0.23) | 30.00 ( 0.74) |
> | P-PT | 0 | 10.93 ( 0.25) | 16.66  (0.30) |
> | SGPT | 5.50 ( 0.37) | 18.91  (0.43) | 27.27 (0.32) |
> | FedPR | 0 | 3.54 (0.14) | 8.62 (0.39) |
> | pFedPG | 0 | 0 | 12.54  (0.41) |
> | **PEP-FedPT (Ours)** | **20.00  (0.38)** | **33.00  (0.50)** | **41.10 (0.48)** |
>
> *The above table is added in sec A.4.8 (pages 22-23)*
>
> **Held-out evaluation vs. baseline applicability (pFedPG)**
>
> The Testing Accuracy reported on held-out clients in our paper is measured strictly under a zero-adaptation protocol. The intent of this evaluation is to assess the generalization ability of the algorithm on unseen clients, without relying on any additional adaptation or personalization steps. This protocol ensures a fair and consistent comparison across methods in the zero-shot setting. Regarding pFedPG, it is important to note that this method is primarily designed for personalized federated learning, where the goal is to adapt models to each client’s local data distribution. It is not inherently intended for zero-shot evaluation on unseen clients. Consequently, its reported performance in our held-out evaluation is extremely low. This outcome should not be interpreted as a failure of pFedPG in general, but rather as a result of applying it outside its intended setting. We included pFedPG in our experiments to illustrate the behavior of personalization-based methods in a strict zero-adaptation regime. By marking the corresponding results as Not Applicable (NA) in the updated draft, we explicitly indicate that the reported numbers do not reflect the method’s standard usage.
>
>
> **On the sensitivity to $\tau$**
>
>  Temperature | CIFAR-100 Accuracy (%) | Tiny-ImageNet Accuracy (%) |
> |------------|----------------------|---------------------------|
> | 1.00E-05   | 82.96  (0.31)              | 78.50 (0.18)                    |
> | 1.00E-04   | 84.03    (0.1)           | 78.80        (0.23)             |
> | 1.00E-03   | 86.24    (0.03)            | 79.86   (0.11)                  |
> | 1.00E-02   | 88.83    (0.14)            | 83.55    (0.19)                 |
> | 1          | 88.62 (0.12)                | 82.81     (0.26)                |
> | 10         | 88.16  (0.05)              | 82.08      (0.15)               |
> | 100        | 87.90   (0.03)             | 81.82     (0.1)                |
>
> This table shows the effect of the temperature parameter, $\tau$, on model accuracy for two datasets: CIFAR-100 and Tiny-ImageNet with $Dir(0.3)$. Accuracy is measured for different values of $\tau$ ranging from $10^{-5}$ to $100$. For both datasets, increasing $\tau$ initially improves accuracy, reaching a peak at the same $\tau$), after which further increases in $\tau$ lead to a drop in performance. This indicates that a moderate temperature helps optimize model performance, while very small or very large temperatures can reduce accuracy.
>
> *The figure corresponding to the above table is added in the Section A.5.6 (pages 25-26)*

---

> ### Author Response · Authors · 2026-01-30
> **Response to Reviewer 72NJ (2)**
>
> **Impact of temeperature on the best layer choices**
>
>
> | Layers     | $\tau$ = 0.0001       | $\tau$ = 0.05        | $\tau$ = 100        |
> |-----------|-----------------|----------------|----------------|
> | 1, 2, 3   | 86.18  (0.05)    | 86.70 (0.04)   | 86.32 (0.16)   |
> | 5, 6, 7   | 86.24  (0.14)    | 88.75 (0.25)   | 87.90 (0.02)   |
> | 9, 10, 11 | 87.77  (0.04)    | 85.68 (0.34)   | 87.35 (0.18)   |
>
>
> Here we show how the accuracy varies on CIFAR-100 under Dirichlet setting. At a low temperature setting ($\tau = 0.0001$), the similarity scores computed from the \texttt{cls} token representations receive a significantly higher relative weight. In this regime, performance is primarily driven by the quality of these scores rather than the depth at which CCMP is injected. Since \texttt{cls} token representations at later layers (e.g., layers 9-11) are more expressive, they provide more accurate similarity estimates, resulting in improved accuracy. As the temperature increases ($\tau = 0.05$ and $\tau = 100$), the influence of the similarity scores is reduced, and the depth of prompt insertion becomes a critical factor. At higher temperatures, sufficient insertion depth is required to enable the prompts to learn meaningful representations, and shallow insertion is no longer adequate to achieve strong performance
>
> We would like to point out that in section A.6.4 in the Appendix, we have performed the theoretical analysis which does not require this particular assumption on the temperature parameter. In this analysis the soft weights are a function of both the data point and the class prior. We still achieve the quadratic upper bound on the loss function under the Lipschitz smoothness assumption at a per-sample level.
>
> **Varying the Dirichlet heterogeneity parameter**
>
> While the main text presents results on CIFAR-100 and Tiny-ImageNet under Dirichlet label-heterogeneity with Dir(0.3), our study explores a broad range of data heterogeneity settings across multiple datasets:
>
> CIFAR-100: Dir(0.3) (pathological non-i.i.d.)
>
> Tiny-ImageNet: Dir(0.3) (pathological non-i.i.d).
>
> DomainNet and iNaturalist: additional heterogeneity settings across domains and client distributions
>
> In total, we evaluate our method under six different heterogeneity scenarios, covering both pathological and moderate data distribution shifts. Hence, we have already explored a diverse set of heterogeneity conditions.
>
>  We now include additional results for CIFAR-100 with Dir(0.1) and Dir(0.5).
>
> **Table: CIFAR results under different Dirichlet partitions**
>
>
> | Method               | Dir(0.1) Mean Acc | Dir(0.1) Worst Acc | Dir(0.5) Mean Acc | Dir(0.5) Worst Acc |
> |----------------------|-----------------|------------------|-----------------|------------------|
> | Head                 | 79.21 ( 0.12)    | 65.48 (0.02     | 79.95 (0.14)    | 72.24  (0.45)     |
> | VPT                  | 84.13 (0.03)    | 71.80  (0.01     | 84.97 (0.01)    | 75.63  (0.01)     |
> | VPT-D                | 87.08 ( 0.02)    | 74.00 (0.02     | **87.92 (0.74)** | 79.00 (0.12)     |
> | P-PT                 | 79.16 ( 0.24 )   | 67.24 (1.24     | 78.80 ( 0.38)    | 64.54 ( 0.68)     |
> | SGPT                 | 85.36 ( 0.01)    | 73.00 (0.16     | 86.04 (0.02)    | 73.49 ( 0.01)     |
> | FedPR                | 81.64 ( 0.32)    | 65.08 (1.80     | 82.11 (0.59)    | 71.42 (0.84)     |
> | pFedPG               | 84.14 ( 0.49)    | 73.62 (0.78     | 73.90 (0.38)    | 60.00 ( 0.98)     |
> | **PEP-FedPT (Ours)** | **90.85 (0.13)** | **84.28 (1.15)** | 87.75 ± 0.04    | **79.51 ± 0.16** |
>
>
> *We have added this result in Table.16 under the sec A.5.5 of the Appendix.*
>
> It can be seen that performance gain is higher when data heterogeneity is more. Our method becomes FedVPT in the iid setup as the scores used to mix class prompts will become identical.
>
> **personalization and generalization trade-off plot**
>
> We have added Personalization and Generalization Trade-off  Sec A.5.8 (26-27)
>
> **Make the early overview more intuitive / higher-level**
>
> We have updated the introduction accordingly.
>
> **Highevel pseudo code**
>
> We have added in the pseudo code in the Algorithm 1 under Sec.4
>
> **Expanded related work**
>
> We have accommodated the mentioned related work in the related work section.

---

> > ### Author Response · Authors · 2026-01-30
> > **Response to Reviewer 72NJ (3)**
> >
> > **Changes Summary**
> >
> > We have colored all the changes requested by reviewer 72NJ  in magenta
> >
> > Inaturalist metric clarification is provided in page (10), additional experiments in sec A.4.8 (pages 22-23)
> >
> > Clarification of pFedpG baseline low accuracy is in pages (10-11)
> >
> > Improved explanation of the mechanism overview in the introduction on page (2)
> >
> > Expanded limitations on page 12
> >
> > Robustness to varying Dirichlet concentration in Sec A.5.5 (pages 25-26)
> >
> > Sensitivity to temperature  is in Section A.5.6 (pages 25-26)
> >
> > Impact of temperature on prompt insertion in Sec A.5.7 (26)
> >
> > Personalization and Generalization Trade-off  Sec A.5.8 (26-27)

---

### Comment · Action_Editor_n21j · 2025-11-17
**Delays to reviewer shortages**

Dear authors,

I'm checking regularly for available reviewers to assign to the paper but there haven't been many available with relevant expertise. Unfortunately, this will delay the review process, most likely by a couple of weeks. I'll keep checking for reviewers and assign them as soon as they are available.

Your AE

---

> ### Author Response · Authors · 2025-11-17
>
> Dear AE,
>
> We thank you very much for letting us know and for your efforts in identifying appropriate reviewers.
>
> Best regards,
> Authors

---

### Author Response · Authors · 2026-02-12

Dear Reviewers

We would like to sincerely thank the reviewers for their time and the expertise they brought to this submission. We believe we have addressed all queries in the revised manuscript (highlighted by reviewer-specific color coding), and we remain fully available to provide further clarification or answer any additional questions.

Regards

Authors

---

### Author Response · Authors · 2026-03-02
**Camera Ready Version**

Dear AE

We have uploaded the camera-ready version with the suggested changes regarding the references, figures, and equations. We would like to sincerely thank you and the reviewers for their constructive feedback and the evaluation. The insights provided helped in refining the work. We appreciate the time and effort dedicated to this review.

Regards

Authors

---

> ### Comment · Action_Editor_n21j · 2026-03-04
> **Missing first author name?**
>
> Dear authors,
>
> The first name of the first author in the paper is simply " M" (with an extra space at the beginning). Is that correct?

---

> > ### Author Response · Authors · 2026-03-05
> >
> > Dear AE,
> >
> > Apologies for the extra space. Thank you for bringing this to our attention. We confirm that the first author's first name is "M". We have removed the leading space and corrected the formatting in the latest camera-ready version.
> >
> > Regards,
> >
> > Authors.

---

### Decision · Action_Editor_n21j · 2026-02-09

**Recommendation:** Accept as is

**Additional Comments:**

I'd like the authors to implement the following changes for the camera-ready version:
1. Reformat the plots. x-axis and y-axis labels are too small in Figures 2, 3, 6, 7, 8, 9, 10, 11, and 13. The style is inconsistent too, for instance Figure 3 uses bold labels, while Figure 2 doesn't.
2. Fix punctuation in equations. At the moment, almost all equations end with no punctuation.
3. Fix references.
i. "W. Deng, C. Thrampoulidis, and X. Li." is the only reference with first names shortened, please make it consistent.
ii. "Cynthia Dwork, Aaron Roth, et al." there should be no "et al." as there are no other authors.
iii. Please add "." after middle names, e.g. "Theodora S Brisimi" should be "Theodora S. Brisimi".

**Audience:**

Yes

**Audience Explanation:**

All reviewers agreed that the paper addresses an important and clearly formulated problem at the intersection of several domains, thus being of interest to a wide audience. In particular, it should be of interest to people working on federated learning and

**Claims And Evidence:**

Yes

**Claims Explanation:**

The reviewers unanimously gave a positive answer. The proposed PEP-FedPT was praised for both its strong empirical performance, and the theoretical justification.